# Multi-gradient permutation survival analysis identifies mitosis and immune signatures steadily associated with cancer patient prognosis

**Xinlei Cai[1], Yi Ye[2], Xiaoping Liu[1], Zhaoyuan Fang[3,4], Luonan Chen[1,2,5], Fei Li[6]\*, Hongbin Ji[1,2,5]\***

[1]Key Laboratory of Systems Health Science of Zhejiang Province, School of Life Science, Hangzhou Institute for Advanced Study, University of Chinese Academy of Sciences, Hangzhou, China; [2]State Key Laboratory of Cell Biology, Shanghai Institute of Biochemistry and Cell Biology, Center for Excellence in Molecular Cell Science, Chinese Academy of Sciences, Shanghai, China; [3]Zhejiang University-University of Edinburgh Institute, Zhejiang University School of Medicine, Haining, China; [4]The Second Affiliated Hospital, Zhejiang University School of Medicine, Hangzhou, China; [5]School of Life Science and Technology, ShanghaiTech University, Shanghai, China; [6]Department of Pathology and Frontier Innovation Center, School of Basic Medical Sciences, Fudan University, Shanghai, China

**\*For correspondence:**
li_fei@fudan.edu.cn (FL);
hbji@sibcb.ac.cn (HJ)

**Competing interest:** The authors declare that no competing interests exist.

## eLife Assessment

This paper contains **valuable** ideas for methodology concerned with the identification of genes associated with disease prognosis in a broad range of cancers. However, there are concerns that the statistical properties of MEMORY are **incompletely** investigated and described. Further, more precise details about the implementation of the method would increase the replicability of the findings by other researchers.

**Abstract** The inconsistency of the association between genes and cancer prognosis is often attributed to many variables that contribute to patient survival. Whether there exist the Genes Steadily Associated with Prognosis (GEARs) and functions remains largely elusive. Here, we developed a novel method named 'Multi-gradient Permutation Survival Analysis' (MEMORY) to screen the GEARs by using RNA-seq data from the TCGA database. We employed a network construction approach to identify hub genes from GEARs and utilized them for cancer classification. In the case of lung adenocarcinoma (LUAD), the GEARs were found to be related to mitosis. Our analysis suggested that LUAD cell lines carrying *PIK3CA* mutations exhibit increased drug resistance. For breast invasive carcinoma (BRCA), the GEARs were related to immunity. Further analysis revealed that *CDH1* mutation might regulate immune infiltration through the EMT process. Moreover, we explored the prognostic relevance of mitosis and immunity through their respective scores and demonstrated it as valuable biomarkers for predicting patient prognosis. In summary, our study offered significant biological insights into GEARs and highlights their potentials as robust prognostic indicators across diverse cancer types.

## Introduction

Cancer is one of the leading causes of death worldwide, with over 200 types identified. Despite major advances in the field, accurately predicting cancer patient prognosis remains a significant challenge (*Tomczak et al., 2015*; *Weinstein et al., 2013*). Previous studies have highlighted the complexity of this task, which is influenced by various factors, including cancer types, clinical stages, therapeutic interventions, nursing care, unexpected comorbidities, and other non-cancer-related illnesses that may interplay (*Liu et al., 2018*; *Smeltzer et al., 2018*; *Miyauchi et al., 2022*; *Teh et al., 2021*). Furthermore, gene expression plays a crucial role in predicting cancer patient prognosis, such as *HER2, VEGF, Ki67*, etc (*Gimotty et al., 2005*; *Wu et al., 2021*; *Mouabbi et al., 2023*; *Kang et al., 2023*; *Cai et al., 2022*; *Zhang et al., 2021*). Nevertheless, there is still a need for further improvement in prognostic accuracy to better inform treatment decisions and patient outcomes.

Survival analysis is commonly used to assess the correlation between genes and cancer patient prognosis (*Lai et al., 2021*). However, inconsistent findings are frequently observed, even within the same type of cancer. For instance, studies exploring the correlation between *CCND1* and prognosis in non-small cell lung cancer (NSCLC) reported contrasting results, including positive correlation, inverse correlation, or negligible influence (*Anton et al., 2000*; *Esposito et al., 2005*; *Dworakowska et al., 2005*). These discrepancies can be attributed to various influential factors, such as differences in sample size, cohort characteristics (including cancer subtypes and tumor staging), and variations in therapeutic approaches (*Freedman, 1982*). Such observations raise an important question: whether there exist Genes Steadily Associated with Prognosis (hereafter referred to as GEARs) that correlate with patient outcomes across different conditions, particularly considering the varying sample sizes used in different studies? Affirmative answers to this question would have significant implications not only for the development of more accurate prognostic models, but also for enhancing our understanding of cancer biology.

The Cancer Genome Atlas (TCGA) is a comprehensive cancer genomics project initiated in the United States (https://portal.gdc.cancer.gov/). It consists of transcriptome data, genomic data, and clinical information pertaining to 33 different cancer types, making it the largest cancer clinical sample database currently available. In this study, we developed a novel method named 'Multigradient Permutation Survival Analysis' (MEMORY) with the utilization of TCGA RNA-seq database, and assessed the potential existence of GEARs across 15 cancer types, each comprising a cohort of over 200 patients. Furthermore, we evaluated the prognostic predictive power of these GEARs and explored their potential biological functions in driving cancer progression.

## Results

### Multi-gradient permutation survival analysis identifies GEARs associated with mitosis and immune across multiple cancer types

GEARs are a group of genes consistently and significantly correlated with cancer patient survival, independent of the sample size. To identify these GEARs, we developed a novel method called 'Multi-gradient Permutation Survival Analysis' (MEMORY). This method allows us to assess the correlation between a specific gene and cancer patient prognosis using available transcriptomic datasets (*Figure 1A*, *Supplementary file 1A*). Initially, we started with a sampling number at 10% of the cohort size, then gradually increased the sampling number with each 10% increment, which was analyzed with 1000 permutations. By calculating the statistical probability of each gene's association with patient survival, we can identify a group of GEARs for further analyses.

In this study, we utilized the TCGA datasets for several reasons, including the diversity of cancer types, the availability of gene expression profiles and patient prognosis information. We set a minimum cohort size of 200 and included 15 eligible cancer types for analysis. These cancer types include bladder urothelial carcinoma (BLCA), breast invasive carcinoma (BRCA), cervical squamous cell carcinoma and endocervical adenocarcinoma (CESC), colon adenocarcinoma (COAD), kidney renal papillary cell carcinoma (KIRC), brain lower grade glioma (LGG), liver hepatocellular carcinoma (LIHC), lung adenocarcinoma (LUAD), lung squamous cell carcinoma (LUSC), ovarian serous cystadenocarcinoma (OV), pancreatic adenocarcinoma (PAAD), prostate adenocarcinoma (PRAD), stomach adenocarcinoma (STAD), thyroid carcinoma (THCA), and uterine corpus endometrial carcinoma (UCEC). As the number of samples increases, the significant probability of certain genes gradually approaches 1.

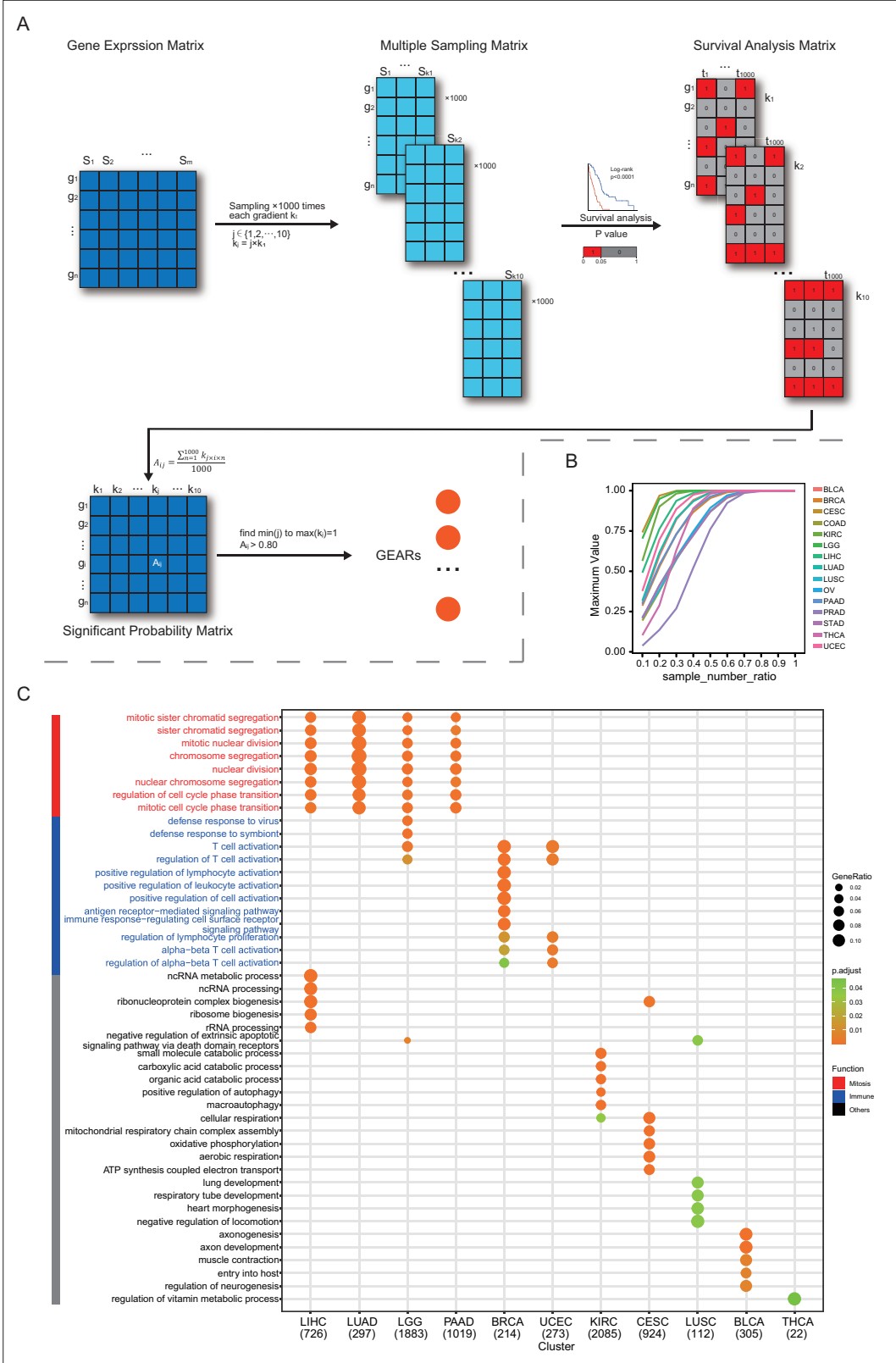

**Figure 1.** MEMORY uncovers the enrichment of mitosis and immune signatures in multiple cancers. (**A**) Algorithm of MEMORY. Sample sizes, ranging from 10% to 100% with 10% intervals ('gradient'), were used for 1000 permutations of survival analyses of all 15 cancer types. Each matrix was divided into high and low expression groups based on median gene expression values, and survival analyses were performed. Log-rank test significance

*Figure 1 continued on next page*

*Figure 1 continued*

results were coded as 1 for significant (p<0.05) and 0 for non-significant (p>0.05) outcomes, forming survival analysis matrices and the summarized significant probability matrix, which allowed the identification of GEARs. (**B**) The maximum of significant probability for each gradient sample size of every cancer type. Sample number rate refers to the percentage of samples in each sampling gradient compared to the total number of samples. (**C**) The pathways were enriched by GOEA based on the GEARs of each cancer type. The displayed pathways represent the top 5 most significant pathways for each cancer type. Mitosis-related pathways were marked in red, whereas immune-related pathways were marked in blue. Only pathways with a Benjamini–Hochberg–adjusted p<0.05 are displayed.

The online version of this article includes the following figure supplement(s) for figure 1:

**Figure supplement 1.** Correlation between sample size and the number of prognostic genes identified by MEMORY across 15 cancer types.

**Figure supplement 2.** The accumulation patterns of significant probability for GEARs across sampling gradients.

---

Once this score reaches 0.8, previously employed as an empirical standard for survival analyses, and remained above this value consistent with further sample gradient increase, the gene is considered a GEAR (*Figure 1B*; *Gebski et al., 2018*). Remarkably, we successfully identified a set of GEARs across all 15 cancer types (*Figure 1—figure supplements 1 and 2*, *Supplementary file 1B*). The GEAR counts in most cancer types ranged from 100 to 1000, with exceptions of CESC, KIRC, LGG, and PAAD with over 1000 GEARs, and THCA with only 22 GEARs (*Supplementary file 1B*). In LUAD, the top 10 GEARs with the highest significance probabilities were *TLE1*, *GNG7*, *ERO1A*, *ANLN*, *DKK1*, *TMEM125*, *S100A16*, *KNL1*, *STEAP1*, and *BEX4*. Most of these genes are known to promote LUAD malignant progression except for S100A16 (*Yao et al., 2014*; *Zheng et al., 2021b*; *Chen et al., 2022b*; *Xu et al., 2019*; *Zhang et al., 2018*; *Wang et al., 2020*; *Huo et al., 2020*; *Zhao et al., 2018*; *Fan et al., 2022*). In other cancer types, *BEX4* was identified as a common GEAR in KIRC, LGG, PAAD, and STAD (*Supplementary file 1B*). *BEX4* is reported as a proto-oncogene promoting cancer onset and malignant progression in multiple cancers, including LUAD, glioblastoma multiforme, and oral squamous cell carcinoma (*Zhao et al., 2018*; *Lee et al., 2021*; *Gao et al., 2016*). We also identified the top 1 GEAR in individual cancer types, such as *TLL1* (BLCA), *PGK1* (BRCA), *RFXANK* (CESC), *DPP7* (COAD), *VWA8* (KIRC), *SCMH1* (LGG), *HILPDA* (LIHC), *TLE1* (LUAD), *CD151* (LUSC), *ANKRD13A* (OV), *SOCS2* (PAAD), *DRG2* (PRAD), *ADAMTS6* (STAD), *PSMB8* (THCA), *ASS1* (UCEC) (*Supplementary file 1C*). Many of these genes were previously reported to be associated with tumorigenesis (*Yao et al., 2014*; *He et al., 2019*; *Matsuura et al., 2017*; *Poplawski et al., 2023*; *Guo et al., 2023*; *Povero et al., 2020*; *Erfani et al., 2021*; *Won et al., 2022*; *Xu et al., 2020*; *Xu et al., 2016*; *Mead, 2022*; *Yang et al., 2018*; *Keshet et al., 2020*). For example, *TLE1* is known as a transcriptional repressor that promotes cell proliferation, migration, and inhibits apoptosis in LUAD (*Palaparti et al., 1997*; *Lin et al., 2023*). Additionally, *PGK1*, a key enzyme in the glycolytic process, has been shown to promote cell proliferation, migration, and invasion in multiple cancers (*Li et al., 2016*; *Fu and Yu, 2020*). These findings demonstrate the intricate link between the functionality of GEARs and the initiation and progression of cancer.

To gain deep insights into the biological functions of these identified GEARs, we conducted Gene Ontology (GO) enrichment analysis (*Figure 1C*). Interestingly, we found the mitosis-related pathways were enriched in LICH, LUAD, LGG, and PAAD, and the immune-related pathways were enriched in BRCA and UCEC (*Waldman et al., 2020*). Additionally, other cancer types exhibited enrichment in various pathways, such as organic acid metabolism pathway in KIRC, oxidative phosphorylation pathway in CESC, organ development-related pathways in LUSC, neurogenesis-related pathways in BLCA, and sustenance metabolism pathways in THCA. Given the crucial role of mitosis in cancer progression and the significance of the immune system in cancer-host interactions, we specifically focused on the mitosis and immune-related GERAs, which accounted for approximately 40% of all the analyzed cancers.

## Identification of hub genes in mitosis and immune-related cancers

To identify crucial genes within the GEARs, we conducted and extracted the higher-ranked edges to construct the core survival network (CSN) (*Figure 2A*, *Supplementary file 1D*). The top 10 GEARs with the highest degree in these networks, selected from those with mean expression >10 TPM, were

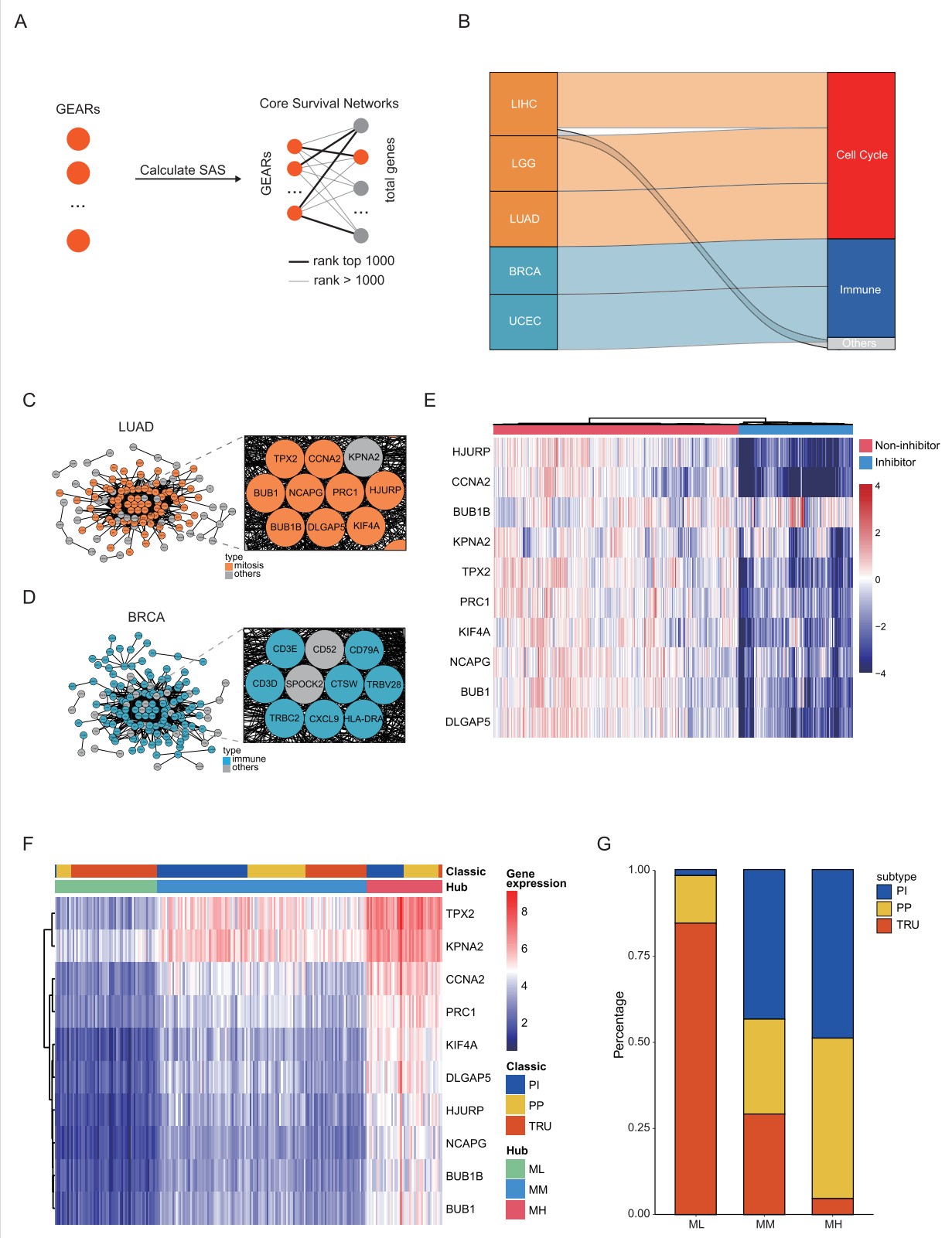

**Figure 2.** Identification of hub genes in mitosis or immune-related cancers. (**A**) GEARs and core survival networks. The SAS of all GEARs was calculated, and CSN was constructed. (**B**) Cancers and their hub gene pathways. The left nodes represent five different cancers, while the right nodes detail the functional types of hub-gene pathways. The height of the edges in the middle represents the proportion of hub-gene pathways corresponding to a specified functional type. Only pathways with a Benjamini–Hochberg–adjusted p<0.05 are displayed. (**C–D**) The CSN of LUAD and BRCA. The

*Figure 2 continued on next page*

*Figure 2 continued*

enlarged section represents the hub gene. (**E**) Heat map showing the hub gene expression of A549 after compounds treatment from cMAP database (*Subramanian et al., 2017*). Blue annotations meant 76 compounds that can inhibit hub gene expression. (**F–G**) LUAD clustering based on hub gene expression in comparison with the classic classification.

The online version of this article includes the following figure supplement(s) for figure 2:

**Figure supplement 1.** CSN and identification of hub genes in 15 cancer types.

**Figure supplement 2.** Hierarchical clustering of 15 cancer types based on hub gene expression.

**Figure supplement 3.** Association between hub gene-based molecular subtypes and clinical stages across pan-cancer.

**Figure supplement 4.** Gene dependency, drug sensitivity screening, and prognostic validation of hub gene subgroups in LUAD.

defined as hub genes (*Zhou et al., 2019*). We then classified the samples using the hub genes derived from these networks and evaluated their clinical relevance (*Figure 2—figure supplements 1–2*). A significant correlation with cancer stages (TNM stages) is observed in most cancer types except for LUSC, THCA, and PAAD (*Figure 2—figure supplement 3*). Furthermore, we conducted GO enrichment analysis on the hub genes selected from CSNs and found that the results were consistent with the GEARs analysis (*Supplementary file 1E*). Specifically, the hub genes in LIHC, LGG, and LUAD were enriched in mitosis-related pathways, whereas the hub genes in BRCA and UCEC were enriched in immune-related pathways (*Figure 2B*). For instance, the nine hub genes in LUAD were associated with functions related to mitosis, whereas the eight hub genes of BRCA were associated with immune-related functions (*Figure 2C–D*).

Next, we conducted survival-dependent analyses on the hub genes of LGG, LIHC, and LUAD. These analyses revealed that mitosis-related hub genes were closely associated with cancer cell viability, especially those hub genes that were correlated with multiple cancer types, such as *CDC20*, *TOP2A*, *BIRC5*, and *TPX2* (*Hwang et al., 1998*; *Uusküla-Reimand and Wilson, 2022*; *Lamers et al., 2011*; *Wittmann et al., 2000*; *Figure 2—figure supplement 4A–C*). The importance of these genes is well established. For instance, *TPX2*, a hub gene in all three cancers, is known to play a crucial role in normal spindle assembly during mitosis and is essential for cell proliferation (*Kufer et al., 2002*). The significance of these hub genes for the survival of cancer cells suggests that the expression levels of these hub genes could be used for inhibitors screening of tumor growth. By integrating the Genomics of Drug Sensitivity in Cancer (GDSC) and Connectivity Map (cMAP) databases, we identified a series of compounds that were able to effectively suppress the expression of the 10 hub genes in LUAD cell lines (*Figure 2E*, *Figure 2—figure supplement 4D–E*). These compounds also significantly inhibited LUAD cell growth and might serve as potential therapeutic agents for LUAD.

We then focused our analysis on the LUAD dataset, which had a larger sample size compared to LGG and LIHC. According to the expression of the hub genes, we classified the LUAD samples into three subgroups: mitosis low (ML), mitosis medium (MM), and mitosis high (MH; *Figure 2F*, *Supplementary file 1F*). A previous study has reported three classic subgroups of LUAD known as the terminal respiratory unit (TRU), the proximal-proliferative (PP), and the proximal-inflammatory (PI) subgroups (*Cancer Genome Atlas Research, 2014*). Our analysis revealed that the ML subgroup was primarily enriched with the TRU subgroup and a small number of samples from the PP subgroup, but not the PI subgroup. The MH subgroup showed high expression of mitosis-related genes and predominantly encompassed PI and PP subgroups, whereas the MM group exhibited intermediate levels of mitosis-related gene expression (*Figure 2F–G*). This suggests that the new categorization method may help identify new factors that influence patient prognosis (*Figure 2—figure supplement 2F-G*).

## Distinct genetic mutation landscapes characterize different clusters of LUAD

The expression of hub genes provided crucial indicators for distinguishing various subgroups of LUAD. However, the underlying mechanisms driving these variations in expression patterns remain unknown. In our subsequent research, we aimed to explore the genomic differences, particularly in terms of gene mutations, among different LUAD subgroups. Initially, we analyzed the variations of genetic mutation landscapes of different LUAD subgroups by assessing the tumor mutation burden (TMB). Interestingly, we found significant differences in TMB among the various subgroups of LUAD (*Figure 3A*). Common driver genes were heterogeneously distributed across the three molecular

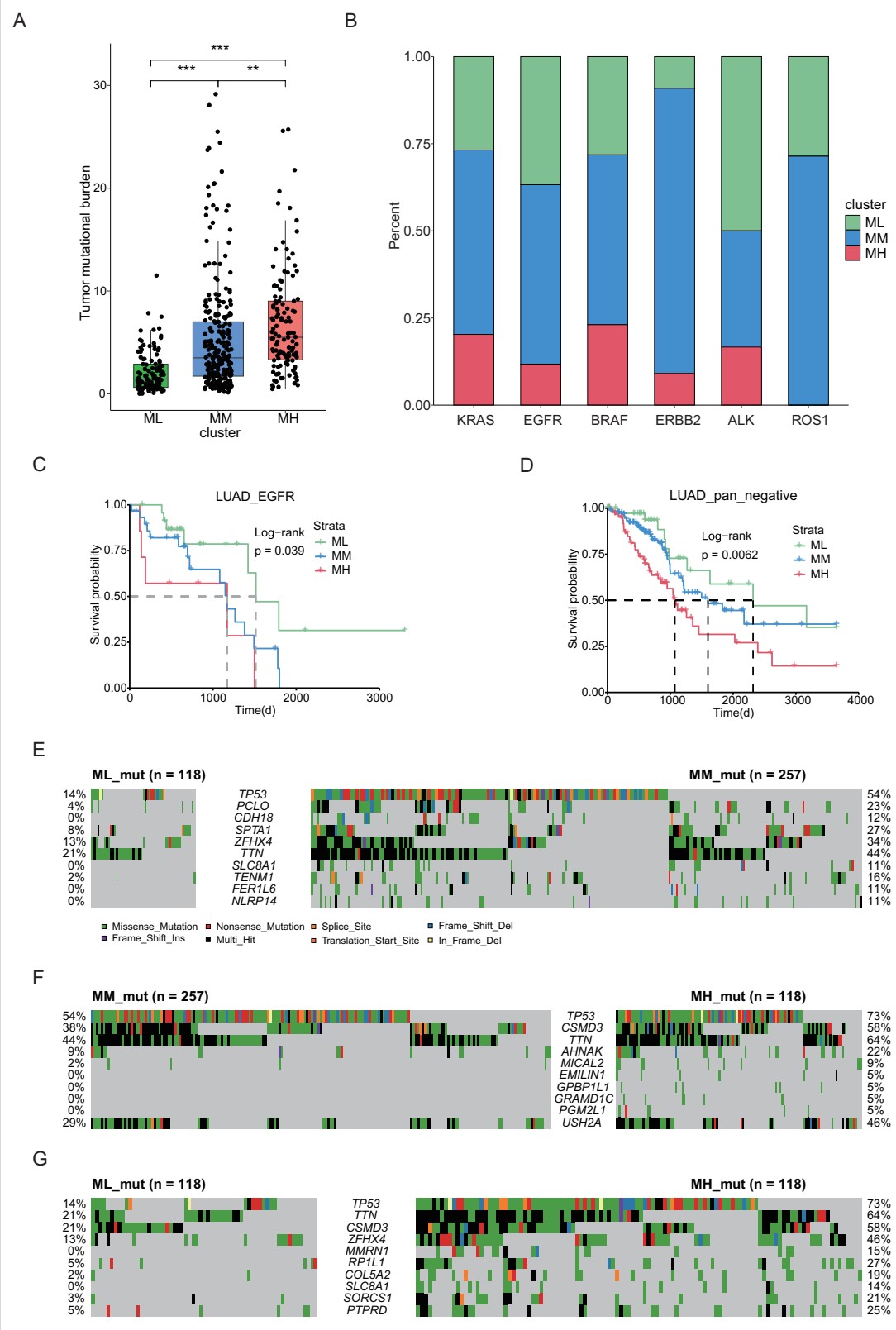

**Figure 3.** Different LUAD subgroups were characterized by unique genetic mutation profiles. (**A**) TMB analysis in three groups of LUAD samples. ***p<0.0005. (**B**) Proportion of different oncogenic drivers, including *KRAS*, *EGFR*, *BRAF*, and *ERBB2* mutations, and *ALK* and *ROS1* fusions in three LUAD subgroups. (**C**) Kaplan-Meier overall survival curves for EGFR-mutation samples of LUAD by hub gene subgroups. (**D**) Kaplan-Meier overall survival curves for pan-negative LUAD samples by hub gene subgroups. (**E–G**) Comparison of top gene mutations in three LUAD clusters, including ML vs. MM (**C**), MM vs. MH (**D**), and ML vs MH (**E**).

subgroups. *ALK* and *ROS1* fusions were most enriched in the prognosis-favorable ML and MM clusters, while *KRAS*, *EGFR*, *BRAF*, and *ERBB2* mutations were more evenly distributed or enriched in the less prognosis-favorable MH group (*Figure 3B*). Due to limited counts for some fusion events, formal statistical comparison was not performed, but the distributional patterns suggested distinct subtype preferences across different driver events. Moreover, low tumor mitotic activity was associated with better prognosis in LUAD subgroups with *EGFR* mutations or pan-negative (no oncogenic alteration in genes including *KRAS*, *EGFR*, *BRAF*, *ERBB2*, *PIK3CA*, *ALK,* and *ROS1*) but not in those with *KRAS* or *BRAF* mutations (*Figure 3C–D*, *Figure 2—figure supplement 4H–I*). Besides, we observed unique mutation characteristics in each of these three subgroups (*Figure 3E–G*). For example, *TP53* mutation, a prevalent tumor suppressor gene mutation, was frequently observed in the MH subgroup with a mutation rate of 73%, compared to mutation rates of 14% in the ML subgroup and 39% in the MM subgroup. While genes, such as *CSMD3*, *RP1L1,* and *ZFHX4,* exhibited similar trends in mutation frequency across the three subgroups. These findings indicate that substantial genomic differences among these three LUAD subgroups are based on the hub genes.

We identified gene mutations that showed significant changes through gene dependence analysis (*Figure 4A*). To further explore the functional implications of these mutations, we enriched them using a pathway system called Nested Systems in Tumors (NeST; https://idekerlab.ucsd.edu/nest/) (*Zheng et al., 2021a*). The results revealed notable differences in multiple functional pathways, including androgen receptor, cell cycle, *TP53*, *EGFR*, IL-6, and *PIK3CA*-related pathways (*Figure 4B*). To gain further insights into the functional implications of these different mutations, we assessed the gene dependency of cells with these mutations by using the DepMap database. Importantly, we observed that the pathway differences between the MM and MH clusters were primarily associated with *PIK3CA*-related pathway. Previous studies have reported that *PIK3CA* mutation confers resistance in colorectal cancer, lung cancer, and breast cancer (*Wang et al., 2018*; *Shibata et al., 2009*; *Hanker et al., 2013*). Therefore, we aimed to validate the role of *PIK3CA* mutations in drug resistance by using public databases. We analyzed two lung adenocarcinoma cell lines, A549 cells and SW1573, harboring *KRAS* mutation and both *KRAS* and *PIK3CA* mutations, respectively, and identified five potentially effective inhibitors (BMS-345541, Dactinomycin, Epirubicin, Irinotecan, and Topotecan) from the previously screened set of 76 LUAD cell growth inhibitors by using GDSC data. Strikingly, SW1573 cells exhibited increased resistance to all these five inhibitors when compared to A549 cells (*Figure 4C*). In line with this, clinical data supported the notion that concurrent *PIK3CA* mutation is a poor prognostic factor for LUAD patients (*Eng et al., 2015*). These findings suggest that *PIK3CA* mutation may contribute to drug resistance in LUAD.

## Distinct clusters of BRCA exhibit different immune infiltration landscapes

In the meantime, we conducted a comprehensive analysis on BRCA, which has the largest sample size among immune-related cancers (*Figure 1C*). We hierarchically grouped the BRCA into three distinct immune subgroups based on the expression of hub genes: immune low (IL), immune medium (IM), and immune high (IH; *Figure 5A*, *Supplementary file 1G*). Interestingly, each PAM50 subtype included tumors from the IL, IM, and IH immune subgroups, with only minor differences in their relative frequencies (*Thorsson et al., 2018*; *Figure 5B*, *Figure 5—figure supplement 1A-B*). Next, we investigated the relationship among PAM50 classification, immune subtypes, and patient prognosis. Our data revealed significant prognostic differences among immune subgroups of the luminal B subtypes but not the luminal A, basal, or HER2 subtypes (*Figure 5—figure supplement 1C–F*), highlighting that integrating our method with traditional classification may enable a more detailed stratification of breast cancer samples. Mutation analysis showed that distinct patterns of gene mutations among the three subgroups had significant differences in mutation frequencies of genes such as *TP53*, *CDH1*, and *LRP2* (*Figure 5C–E*). We conducted a comparison of the proportions of immune cells across these three subgroups and observed a strong association between the overall immune cell proportion and the clustering results based on hub genes (*Figure 5F*, *Figure 5—figure supplement 2A–J*). Specifically, the IH subgroup exhibited significantly higher proportions of CD8[+] T cells and Treg cells, with median percentages at 3.9% and 5.5%, respectively, compared to 1.2% and 2.8% in the IM subgroup, and 0.2% and 1.6% in the IL subgroup. Together, these findings suggest that distinct immune responses across the three subgroups potentially link genetic mutation patterns to differences in the immune microenvironment.

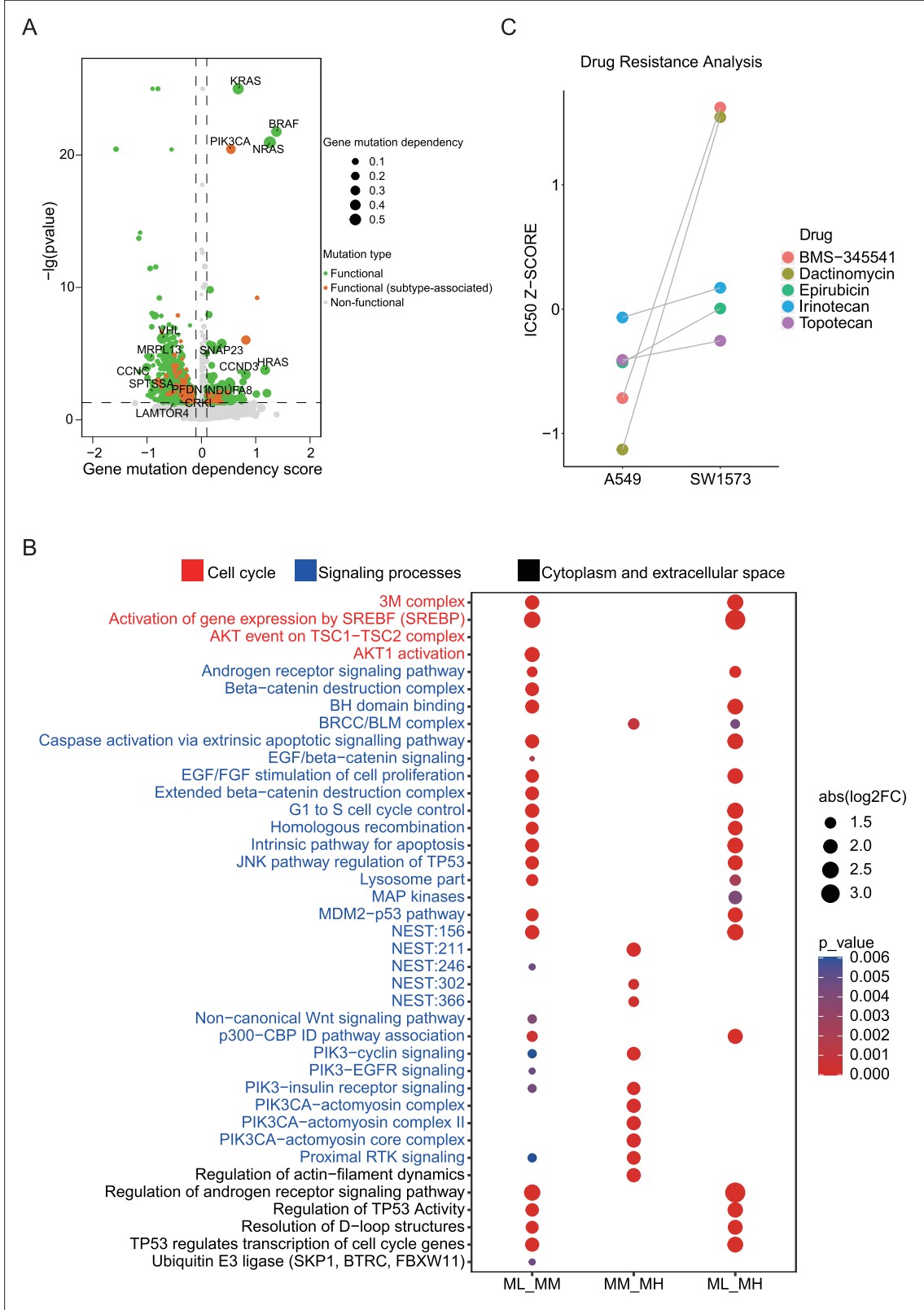

**Figure 4.** PIK3CA mutation associates with mitosis and drug resistance in LUAD. (**A**) Cancer-related gene mutations were annotated based on gene dependency scores data from Depmap database. Functional mutations were indicated in green; functional (subtype-associated) mutations were highlighted in orange; non-functional mutations were indicated in gray. (**B**) The analysis of the NeST differential pathways across three groups including ML vs. MM, MM vs. MH, ML vs MH. Only pathways with a Benjamini–Hochberg–adjusted $p<0.05$ are displayed. (**C**) Comparison of the IC50 z-score of five tumor cell growth inhibitors for A549 (*KRAS* mutation) and SW1573 (*KRAS* and *PIK3CA* mutation).

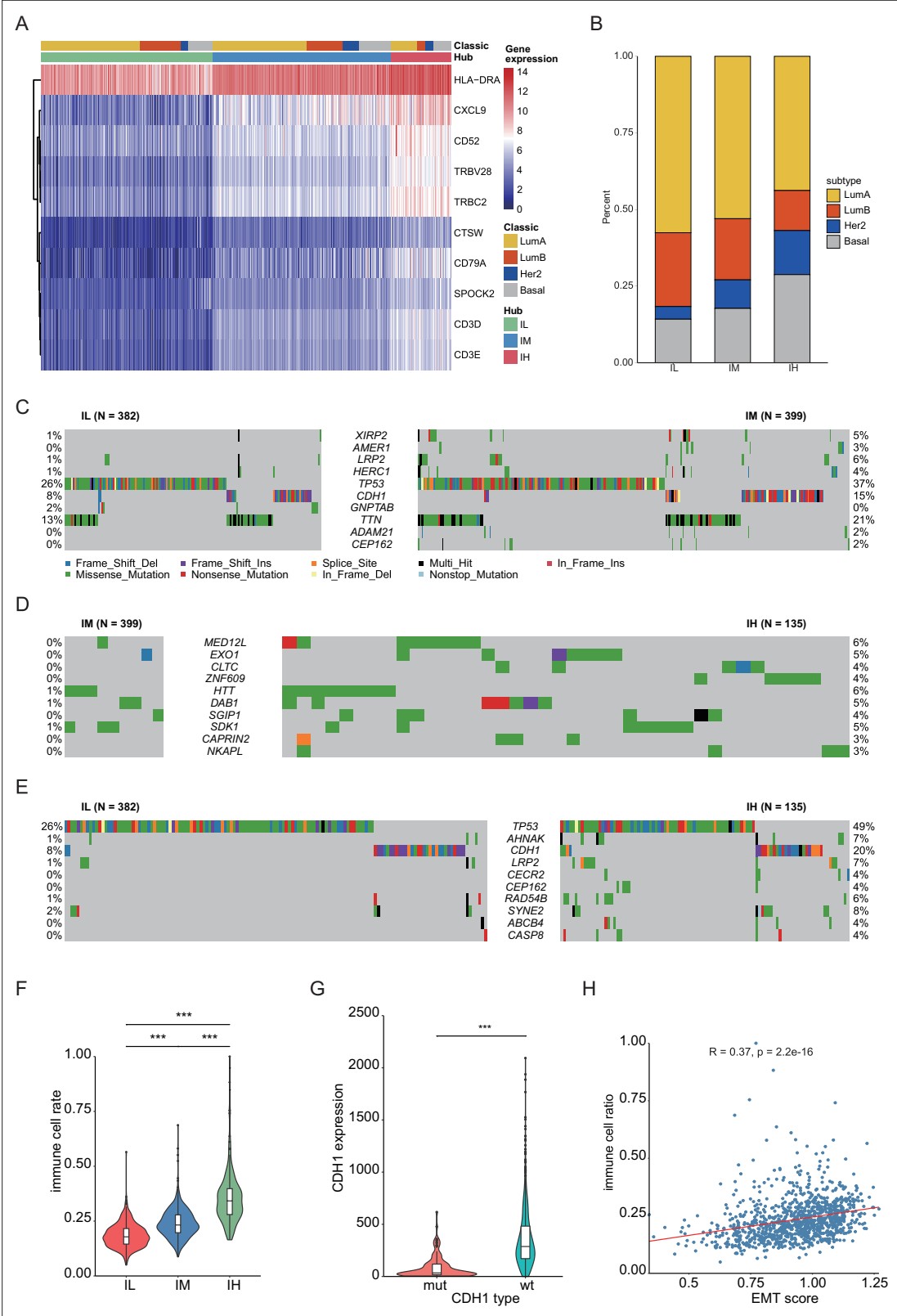

**Figure 5.** The hub gene classification of BRCA revealed a significant association between EMT and immune infiltration. (**A**) Hub gene expression heatmap. The heatmap contains the result of hub gene classification and PAM50 classification of BRCA (***Thorsson et al., 2018***). (**B**) Comparison of molecular classification based on hub genes with PAM50 classification. (**C–E**) Comparison of gene mutations in three groups of BRCA samples, including IL vs. IM (**C**), IM vs. IH (**D**), IL vs. IH (**E**). (**F**) Comparison of the total immune cell rate in three BRCA clusters. ***p<0.0005. (**G**) Comparison of *CDH1*

*Figure 5 continued on next page*

*Figure 5 continued*
expression in BRCA samples with wildtype or mutant *CDH1*. \*\*\*p<0.0005. (**H**) A schematic diagram illustrating the correlation between the EMT score and immune cell rate. The red line represents the fitting curve. The correlation analysis was performed using Pearson correlation coefficient. Statistical significance for all pairwise group comparisons was assessed with the Wilcoxon test.

The online version of this article includes the following figure supplement(s) for figure 5:

**Figure supplement 1.** Prognostic analysis of BRCA molecular subtypes based on hub gene signatures.

**Figure supplement 2.** Immune cell infiltration ratio and their correlation with CDH1-mediated EMT in BRCA.

Next, we explored the specific genomic factors influencing immune infiltration in BRCA. Mutation analysis revealed that *CDH1* gene mutation ranked as the top two genetic alteration in BRCA samples, following *TP53* mutation (*Figure 5C–E*). A previous study has reported that *CDH1* is involved in the mechanisms regulating cell-cell adhesions, mobility, and proliferation of epithelial cells (*Meigs et al., 2002*). We found that *CDH1*-mutant samples exhibited significantly lower *CDH1* expression compared to *CDH1*-wildtype samples, and *CDH1* expression showed a close correlation with immune cell infiltration (*Figure 5G*, *Figure 5—figure supplement 2K*). These results were consistent with clinical observations of *CDH1* mutation and high immune infiltration in invasive lobular carcinoma of the breast (*An et al., 2018*).

The above result encouraged us to investigate how *CDH1* mutation influenced immune infiltration. In BRCA, we observed a significant correlation between the expression of *CDH1* and EMT marker genes *VIM* and *TWIST2* (*Figure 5—figure supplement 2L–M*; *Zeisberg and Neilson, 2009*; *Yang et al., 2004*). This suggests that the regulation of *CDH1* is intricately linked with the EMT process. EMT score analysis showed that there's a positive correlation between the EMT score and the proportion of immune infiltration (*Figure 5H*), which suggests that *CDH1* might influence immune infiltration through the EMT process.

## Mitotic and immune signatures predict patient prognosis at pan-cancer level

Finally, we analyzed the association of *CDH1* and *PIK3CA* with specific biological processes at pan-cancer level. Our data showed that *PIK3CA* level was positively correlated with mitotic scores in most of the cancer types, whereas the correlation between *CDH1* expression and the proportion of immune cell infiltration was positive in PRAD, LGG, OV, Uveal Melanoma (UVM), THCA, LIHC, and LUAD but negative in BRCA, Testicular Germ Cell Tumors (TGCT), Thymoma (THYM), LUSC, BLCA, Head and Neck squamous cell carcinoma (HNSC), Sarcoma (SARC), Esophageal carcinoma (ESCA), PAAD, and STAD (*Figure 6A–B*). Eventually, we sought to explore the prognostic relevance of mitosis and immune to patient outcomes at the pan-cancer level. The scores for these biological processes were computed using RNA-seq data from the TCGA database, and the median score was used as a cutoff to categorize patients into different subgroups. Out of 33 cancer types, 19 showed a statistically significant correlation (p<0.05) with at least one pathway score (*Figure 6C*). Specifically, 10 cancer types were exclusively associated with the mitosis score, 4 cancer types were exclusively associated with the immune score, and 5 cancer types exhibited a correlation with both mitosis and immune scores simultaneously. Overall, the identification of mitosis and immune-related biological processes as significant prognostic factors at the pan-cancer level suggests their potential utility as valuable biomarkers for predicting patient prognosis.

## Discussion

Due to the multifaceted nature of variables affecting cancer patient prognosis, it remains uncertain whether there exists a set of genes steadily associated with cancer prognosis, regardless of sample size and other factors. Here, we utilized the MEMORY method to address this question and discovered that all the cancer types have GEARs. We observed significant variation in the number of GEARs among different cancer types, indicative of cancer type-specific patterns. The substantial heterogeneity in driver genes and mortality rates among various cancer types could potentially explain this phenomenon. For example, THCA, known for its low malignancy, displays the fewest genetic expression alterations among all the studied cancer types. This observation could be correlated with the

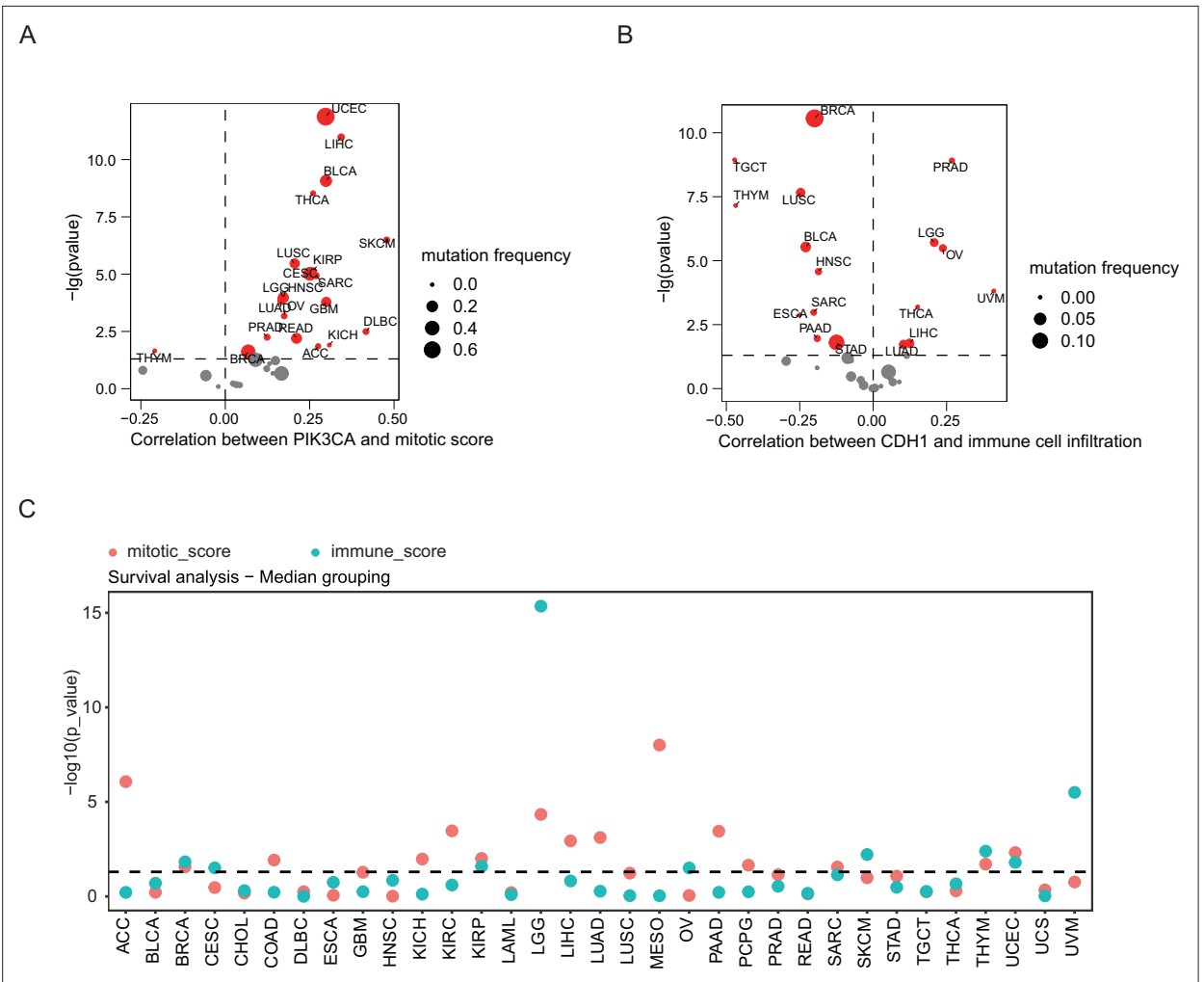

**Figure 6.** Mitosis and immune signatures predict patient prognosis at the pan-cancer level. (**A**) The correlation of mitosis scores of 33 TCGA cancer types with *PIK3CA* expression. (**B**) The correlation of immune cell infiltration rate of 33 TCGA cancer types with *CDH1* expression. (**C**) The mitosis and immune-related pathway scores of 33 cancer types. Median score was utilized as a threshold to categorize patients for survival analysis. Among the 33 cancer types examined, 10 cancer types were exclusively associated with the mitosis score, 2 cancer types were exclusively associated with the immune score, and 5 cancer types showed a correlation with both mitosis and immunity scores concurrently. Dots represent $-log_{10}(p)$ values from log-rank tests comparing high vs. low score groups for each cancer type. The dashed line marks the nominal significance threshold (*P*=0.05).

relatively high five-year survival rate in THCA, which exceeds 50% even in advanced stage (*Egner, 2010*). In contrast, PRAD, despite a favorable prognosis similar to THCA, exhibits a significantly higher number of genetic expression alterations. We hypothesize that the positive prognosis in PRAD cases might be largely attributed to early diagnosis, which enables timely treatment for the majority of PRAD patients (*Rebello et al., 2021*). This discrepancy between the number of genetic alterations and prognosis in THCA and PRAD highlights the intricate nature of cancer genetics and emphasizes the importance of personalized considerations in cancer treatment and prognosis evaluation. Furthermore, certain genes are commonly found in the GEARs of various cancer types, indicating their potential significance in cancer development. For instance, *BEX4* is present in the GEARs of LUAD, KIRC, LGG, PAAD, and STAD and is known to play an oncogenic role in inducing carcinogenic aneuploid transformation via modulating the acetylation of α-tubulin (*Lee et al., 2016*; *Lee et al., 2018*). Aneuploidy is a hallmark characteristic of cancer and can lead to alterations in the dosage of onco-genes or tumor suppressor genes, thereby influencing tumor initiation and progression (*Beroukhim et al., 2010*; *Davoli et al., 2013*; *Girish et al., 2023*; *Bosco et al., 2023*). The regulation of aneu-ploid transformation by *BEX4* may represent a common mechanism through which this gene impacts prognosis across different cancer types. Subsequently, we discovered that GEARs across different

cancers displayed distinct functional characteristics. Notably, a recurrent theme was the prominence of mitosis-related and immune-related features. Specifically, the GEARs of LGG, LIHC, and LUAD were enriched in mitosis-related pathways, whereas BRCA and UCEC showed the enrichment of immune-related pathways. Mitosis and immune processes have been widely recognized as having a significant impact on patient prognosis (*Ha et al., 2016*; *Zhou et al., 2018*; *Mäkinen et al., 2017*; *Byrne et al., 2020*; *Liu et al., 2021*). However, current clinical guidelines do not yet recommend the use of transcriptomic data to assess scores related to mitotic and immune pathways for predicting patient outcomes, despite the well-established association between these pathways and prognosis in cancer (*Ducreux et al., 2015*; *Cardoso et al., 2019*; *Oaknin et al., 2022*; *Vogel et al., 2018*; *Stupp et al., 2014*; *Postmus et al., 2017*). Cancers enriched with mitosis pathways often exhibit heterogeneous tumor growth kinetics across individuals, with tumor size being one of the crucial factors influencing patient survival (*Gui et al., 2018*; *Alvarez et al., 2022*; *Infante et al., 2013*). For instance, in the case of LGG, the growth rate of tumors directly impacts the patient's prognosis due to the primary location in the brain. Consequently, alterations in the expression levels of genes related to mitosis are often indicative of the prognosis of LGG (*Hoshino, 1984*). Cancer types that are enriched in immune-related pathways, such as BRCA and UCEC, are closely associated with deficiencies in DNA mismatch repair (MMR; *Sajjadi et al., 2021*; *Doghri et al., 2019*). Cancers enriched in immune-related pathways often exhibit heterogeneous tumor growth kinetics across individuals, where tumor size is closely correlated with patient survival.

To explore the underlying differences between samples associated with distinct mitotic and immune-related pathways, we employed GEAR analysis to identify hub genes for further molecular classification and mutation analysis. Specifically, we focused on LUAD and BRCA, two representative cancer types exhibiting enriched mitosis and immune signatures in GEARs. By classifying the hub gene, we divided LUAD into three subgroups: ML, MM, and MH, which displayed significant differences in survival outcomes. Subsequently, we utilized NeST to analyze the mutations within these subgroups. Notably, there were significant differences observed in cell cycle and signal transduction-related pathways among ML, MM, and MH subgroups. Of particular importance, the *PIK3CA*-related pathway emerged as a key differentiating pathway between the MH and MM subgroups. Our analyses suggest that LUAD cell lines carrying *PIK3CA* mutations may exhibit increased drug resistance. This aligns with previous studies demonstrating that targeting the PI3K pathway can overcome drug resistance (*Zhang et al., 2016*; *Donev et al., 2011*). Of course, future research is warranted to elucidate the exact functional role of *PIK3CA* mutations in LUAD.

To investigate the factors influencing immune infiltration in BRCA, we classified the samples based on hub genes and analyzed the differentially mutated genes. Interestingly, we found that *CDH1* mutation occurred at a high frequency in the IH subgroups. This led us to speculate that *CDH1* plays a crucial role in the regulation of immune infiltration in BRCA. Furthermore, previous studies have reported that *CDH1* inactivation promotes immune infiltration in breast cancer (*An et al., 2018*). *CDH1* is a vital gene associated with EMT, and various studies have demonstrated the close relationship between EMT and the tumor immune microenvironment in different cancers (*Wang et al., 2021*). Consistently, our results also supported the correlation between EMT score and immune infiltration in BRCA. These findings suggest that the mutations in *PIK3CA* and *CDH1*, identified through GEAR analysis, have significant impacts on cancer development and hold potential value in improving clinical therapies. This further emphasizes the importance of GEARs in understanding cancer biology and guiding treatment strategies.

Lastly, we investigated the prognostic predictive capabilities of the mitosis score and immune score at the pan-cancer level. Surprisingly, we found that approximately half of the cancer types exhibited significant correlations between these two scores and patient prognosis. Interestingly, even for cancers originating from the same primary location, their correlations with these scores could differ, indicating the potential diversity of mechanisms underlying cancer-related mortality. For instance, both LGG and GBM are brain tumors, but the primary risk factor for patient prognosis in LGG is closely linked to tumor diameter, whereas the main risk factor for GBM patients lies in its high invasiveness and challenge of surgical resection (*Brown et al., 2019*; *Chen et al., 2022a*).

Although the present study defined a preliminary catalog of GEARs and supported their relevance with extensive biological data, we recognize several outstanding methodological limitations at both the algorithmic and analytical levels. Chief among them is the need for rigorous, large-scale

multiple-testing adjustment before any GEAR list can be considered clinically actionable. Because this work was conceived as an exploratory screen, such correction was intentionally deferred; nonetheless, forthcoming versions of MEMORY will incorporate a dedicated false-positive–control module that will be applied to the consolidated GEAR catalogue prior to any translational use. Although GEARs show robust prognostic associations, the CSN edges are undirected, and hub genes serve primarily as stable biomarkers. Future work will explore causality and therapeutic implications. As a result, the hub genes identified from GEAR in the CSN may primarily serve as stable and effective biological markers. Additionally, through multi-omics analysis, we obtained some functional mutations, but the therapeutic significance of these mutations remains to be elucidated. For example, further study is needed to understand the exact role of *PIK3CA* mutations in promoting tumor cell proliferation and drug resistance. Similarly, the association of *CDH1* mutations with the infiltration of multiple immune cell types also warrants additional experimental investigation. In our future studies, we plan to utilize protein-protein interaction networks and pathway databases to construct a novel network based on the CSN. This approach will allow us to directly screen genes from GEARs that could potentially serve as therapeutic targets. Undoubtedly, future efforts are still required to utilize protein-protein interaction networks and pathway databases to construct a new network based on CSN.

In conclusion, our study utilized the MEMORY algorithm to identify GEARs in 15 cancer types, highlighting the significance of mitosis and immunity in cancer prognosis. Our findings demonstrate that GEARs possess substantial biological significance beyond their role as prognostic biomarkers. This study provides valuable guidance for establishing standards for survival analysis evaluation and holds potential for the development of novel therapeutic strategies.

## Methods

### Datasets

The gene expression profiles and clinical information of 33 cancers were obtained from the TCGA database and downloaded by the GDC data website (https://portal.gdc.cancer.gov/). All gene expression data and survival data were integrated and normalized.

### Multi-gradient permutation survival analysis

RNA-seq count data were normalized by TPM from the gene expression matrix (genes × samples) for each cancer type. We randomly sampled the gene expression data according to the gradient. The sampling strategy was as follows: Ten gradient increases in sample size were pre-set, ranging from approximately 10% to about 100%, with intervals of 10%. Random samples were taken from total samples of each cancer 1000 times, based on each pre-set sample size. A multiple sampling matrix was obtained after the sampling strategy was performed in each gradient. Then survival analysis was performed using the R package 'survival'. Gene expression values were dichotomized based on the median expression level of each gene in the sampled matrix. The survival analysis method was as follows according to the median expression value of every gene, every sampling matrix was divided into a high and low expression group. This approach divides the samples into high and low expression groups with equal size, providing a standardized grouping strategy for survival analysis. We used 1 for significant survival analysis results (p<0.05) and 0 for non-significant results (p>0.05). The survival analysis matrices were obtained after these processes. This generated a binary matrix of shape (genes ×1000) per gradient, representing the significance profile across permutations. The binary matrix was integrated to a significant-probability matrix by a formula:

$$A_{ij} = \frac{\sum_{n=1}^{1000} k_{j \times i \times n}}{1000} \tag{1}$$

where the $A_{ij}$ is the value from row $i$ and column $j$ in the significant-probability matrix. We defined the sampling size $k_j$ reached saturation when the max value of column j was equal to 1 in a significant-probability matrix. The least value of $k_j$ was selected, and the genes with their corresponding $A_{ij}$ greater than 0.8 were extracted as GEAR.

### Construction of core survival network

We also defined survival analysis similarity (SAS) as the similarity of the effect on patient prognosis in two genes. For each gene, the 1 000 permutation tests performed at the first sampling gradient yield

a binary vector $A = (A_1, \cdots, A_{1000}), B = (B_1, \cdots, B_{1000})$ (1=significant, 0=non-significant). Vectors from the first sampling gradient ($k_1$, approximately 10% of the cohort) were used as input. At this sample size, significance patterns exhibited high variability, with many genes not yet consistently significant. This variability was utilized to enable SAS to differentiate gene pairs with concordant versus discordant behavior.

Let $a = \sum_{i=1}^{1000} A_i$, $b = \sum_{i=1}^{1000} B_i$, $c = \sum_{i=1}^{1000} A_i B_i$ where $c$ is the number of permutations in which both genes are significant and $a$, $b$ are the total significant counts for each gene.

SAS is calculated as

$$SAS(A, B) = \frac{c}{a + b - 2c + 1} \tag{2}$$

a Jaccard-like metric bounded between 0 (no overlap) and 1 (complete overlap); the "+1" term prevents division by zero.

Pair-wise SAS values were computed between every GEAR and all other genes. The 1 000 SAS were selected to construct a core survival network (CSN) in Cytoscape (**Shannon et al., 2003**). Node degrees were calculated, and the top 10 GEARs with the highest degree in these networks, selected from those with mean expression >10 TPM, were defined as hub genes. We constructed the Core Survival Network (CSN) using the top 1000 gene pairs ranked by SAS values. As the CSN was designed as a heuristic similarity network to prioritize genes for exploration, edge selection was based on empirical ranking rather than formal statistical thresholds.

## Gene ontology enrichment analysis

Gene ontology has been used to classify genes based on functions. The gene functions were divided into three types, including molecular function (MF), biological process (BP), and cellular component (CC). ClusterProfiler is an R package for gene set enrichment analysis (**Yu et al., 2012**). The gene ontology (GO) enrichment analysis was processed in GEARs and hub genes by ClusterProfiler.

## Hub gene classification

Tumor samples were genotyped using RNA-seq data. The following steps outline the genotyping and clustering methods used in this study. The expression matrix of hub genes corresponding to various cancer types was extracted from the RNA-seq data. This involved filtering the RNA-seq data to retain only the expression levels of the identified hub genes. To normalize the expression data and reduce the impact of extreme values, a pseudo count of 1 was added to each expression value, and the resulting matrix $M$ was log-transformed using the formula: $M' = log_2(M + 1)$. The log-transformed matrix $M'$ was then subjected to hierarchical clustering. This clustering was performed in R (v4.3) using the Ward's method (hclust, method = "ward.D2") to minimize the variance within clusters. The distance metric used was the Euclidean distance (dist, method = "euclidean"). The number of clusters was set to 3 using the cutree function (cutree(model, k=3)), based on visual inspection of the dendrogram structure.

## Calculation of tumor mutation burden (TMB) and differential mutation

The TMB and differential mutation gene analysis were carried out to explore the differences between different genotyped samples at the genomic level. The data are from the gene mutation data of TCGA tumor samples, and the grouping information is from the hub gene hierarchical clustering results. Maftools is an R package for the analysis of somatic variant data, which can export results in the form of charts and graphs (**Mayakonda et al., 2018**). The calculation of TMB and difference mutation analysis was used by the maftools.

## Quantifying the effect of gene mutations for tumor cell viability

Gene dependence refers to the extent to which genes are essential for cell proliferation and survival. The Cancer Dependency Map (Depmap, https://depmap.org/portal/) database provides genome-wide gene dependence data for a large number of tumor cell lines (**Tsherniak et al., 2017**). Gene mutation data of cell lines was obtained from the CCLE database (https://sites.broadinstitute.org/ccle). The genes appearing in the gene mutation data of cell lines were extracted and sorted into

mutation lists. Each gene in the list was then analyzed for survival-dependent differences. For each gene in the mutation list, we compared the gene dependence score ($S$) between cell lines with and without the gene mutation. Specifically, cell lines from the CCLE mutation dataset harboring a mutation in a given gene were classified into one group, while cell lines without the mutation constituted the control group. $S$ for the mutation ($S_m$) and wild-type ($S_{wt}$) cell lines were then extracted from the DepMap database.

A two-sample t-test was used to assess the statistical significance of differences between $S_m$ and $S_{wt}$. The test yielded a p-value, with an alpha level set at 0.05, to determine the significance of the difference in means. $D_m$ was defined as gene mutation dependence, and the computational formula was as follows:

$$D_m = |\bar{s}_m - \bar{s}_{wt}| \tag{3}$$

Subsequently, we conducted a standardization assessment of $D_m$, employing the following formula:

$$S_{dm} = \frac{2\left(\bar{s}_m - \bar{s}_{wt}\right)}{\bar{s}_m + \bar{s}_{wt}} \tag{4}$$

In this study, we categorized mutations identified in cell lines into three groups based on their impact on gene dependency. Functional mutations refer to the mutations that exhibit significant changes in gene dependency scores, with a p-value <0.05 and $S_{dm}$ >0.1. Non-functional mutations were those without significant changes in gene dependency scores compared to wild type. Subtype-associated functional mutations were a subset of functional mutations that show statistically significant differences in occurrence frequency across different LUAD subtypes.

## Identification of drugs downregulating hub gene expression

The IC50 data of 76 compounds of LUAD cell lines were obtained from Genomics of Drug Sensitivity in Cancer (GDSC) database (*Yang et al., 2013*). The hub gene expression data of A549 cell lines after treatment with 76 compounds was obtained from Connectivity MAP (cMAP, https://clue.io/) database, and the data were grouped by hierarchical clustering (*Subramanian et al., 2017*). The drugs that were presented only in the hub gene expression suppression group were considered to be effective against the LUAD cell.

## Immune infiltration analysis

Immune infiltration data was calculated by quantiseq method (*Finotello et al., 2019*). Immunedeconv was an R package for unified access to computational methods for estimating immune cell fractions from bulk RNA-seq data (*Sturm et al., 2020*). The data were from the gene expression data of TCGA tumor samples after TPM was standardized. Then the immune cell fractions of tumor samples were calculated by the quantiseq method invoking immunedeconv.

## Biological function score calculation

GSVA is an R package for calculating biological function score based on a single sample (*Hänzelmann et al., 2013*). The data was from TCGA in the calculation process, and the software package was GSVA. The immune score was T-cell infiltration rate of that calculated by quantiseq (*Finotello et al., 2019*). The gene sets that calculated the mitosis score were obtained from gene ontology (GO:0140014).

## Acknowledgements

This work was supported by the National Key Research and Development Program of China (grants 2022YFA1103900 to HJ, FL; 2020YFA0803300 to HJ); the National Natural Science Foundation of China (grants 82341002 to HJ, 32293192 to HJ, 82030083 to HJ, 82141101 to FL, 82372794 to FL); the Basic Frontier Scientific Research Program of Chinese Academy of Science (ZDBS-LY-SM006 to HJ); the Innovative research team of high-level local universities in Shanghai (SSMU-ZLCX20180500 to HJ). We thank Gongwei Wu for his helpful comments and suggestions on the manuscript.

# Additional information

## Funding

| Funder | Grant reference number | Author |
|---|---|---|
| National Key Research and Development Program of China | 2022YFA1103900 | Fei Li<br>Hongbin Ji |
| National Key Research and Development Program of China | 2020YFA0803300 | Hongbin Ji |
| National Natural Science Foundation of China | 82341002 | Hongbin Ji |
| National Natural Science Foundation of China | 32293192 | Hongbin Ji |
| National Natural Science Foundation of China | 82030083 | Hongbin Ji |
| National Natural Science Foundation of China | 82141101 | Fei Li |
| National Natural Science Foundation of China | 82372794 | Fei Li |
| Chinese Academy of Sciences | Basic Frontier Scientific Research Program ZDBS-LY-SM006 | Hongbin Ji |
| Innovative research team of high-level local universities in Shanghai | SSMU-ZLCX20180500 | Hongbin Ji |

The funders had no role in study design, data collection and interpretation, or the decision to submit the work for publication.

## Author contributions

Xinlei Cai, Data curation, Formal analysis, Validation, Visualization, Methodology, Writing – original draft; Yi Ye, Investigation, Methodology; Xiaoping Liu, Methodology, Writing – original draft; Zhaoyuan Fang, Writing – review and editing; Luonan Chen, Methodology; Fei Li, Supervision, Funding acquisition, Project administration, Writing – review and editing; Hongbin Ji, Supervision, Funding acquisition, Investigation, Writing – review and editing

## Author ORCIDs

Xinlei Cai ⓘ https://orcid.org/0009-0005-4295-6371
Xiaoping Liu ⓘ https://orcid.org/0000-0002-3246-4227
Hongbin Ji ⓘ https://orcid.org/0000-0003-0891-6390

Reviewer #1 (Public review): https://doi.org/10.7554/eLife.101619.3.sa1
Reviewer #2 (Public review): https://doi.org/10.7554/eLife.101619.3.sa2
Reviewer #4 (Public review): https://doi.org/10.7554/eLife.101619.3.sa3
Author response https://doi.org/10.7554/eLife.101619.3.sa4

# Additional files

## Supplementary files

MDAR checklist

Supplementary file 1. Supplementary tables containing dataset sampling sizes, GEARs, top 10 GEARs, core survival network, hub genes, sample classification, and mitotic signature.

## Data availability

The current manuscript is a computational study, so no data have been generated for this manuscript. Code is available at https://github.com/XinleiCai/MEMORY (copy archived at *Cai, 2024*).

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
