## [Editor Report · eLife Assessment]

This paper contains **valuable** ideas for methodology concerned with the identification of genes associated with disease prognosis in a broad range of cancers. However, there are concerns that the statistical properties of MEMORY are **incompletely** investigated and described. Further, more precise details about the implementation of the method would increase the replicability of the findings by other researchers.

---

## [Referee Report · Reviewer #1 (Public review)]

Summary:

The authors propose a new technique which they name "Multi-gradient Permutation Survival Analysis (MEMORY)" that they use to identify "Genes Steadily Associated with Prognosis (GEARs)" using RNA-seq data from the TCGA database. The contribution of this method is one of the key stated aims of the paper. The majority of the paper focuses on various downstream analyses that make use of the specific GEARs identified by MEMORY to derive biological insights, with a particular focus on lung adenocarcinoma (LUAD) and breast invasive carcinoma (BRCA) which are stated to be representative of other cancers and are observed to have enriched mitosis and immune signatures, respectively. Through the lens of these cancers, these signatures are the focus of significant investigation in the paper.

Strengths:

The approach for MEMORY is well-defined and clearly presented, albeit briefly. This affords statisticians and bioinformaticians the ability to effectively scrutinize the proposed methodology and may lead to further advancements in this field. The scientific aspects of the paper (e.g., the results based on the use of MEMORY and the downstream bioinformatics workflows) are conveyed effectively and in a way that is digestible to an individual that is not deeply steeped in the cancer biology field.

Weaknesses:

Comparatively little of the paper is devoted to the justification of MEMORY (i.e., the authors' method) for identification of genes that are important broadly for the understanding of cancer. The authors' approach is explained in the methods section of the paper, but no comparison or reference is made to any other methods that have been developed for similar purposes, and no results are shown to illustrate the robustness of the proposed method (e.g., is it sensitive to subtle changes in how it is implemented).

For example, in the first part of the MEMORY algorithm, gene expression values are dichotomized at the sample median, and a log-rank test is performed. This would seemingly result in an unnecessary loss of information for detecting an association between gene expression and survival. Moreover, while dichotomizing gene expressions at the median is optimal from an information theory perspective (i.e., it creates equally sized groups), there is no reason to believe that median-dichotomization is correct vis-à-vis the relationship between gene expression and survival. If a gene really matters and expression only differentiates survival more towards the tail of the empirical gene expression distribution, median-dichotomization could dramatically lower power to detect group-wise differences. Notwithstanding this point, the reviewer acknowledges that dichotomization offers a straightforward approach to model gene expression and is widely used. This approach is nonetheless an example of a limitation of the current version of MEMORY that could be addressed to improve the methodology.

If I understand correctly, for each cancer the authors propose to search for the smallest subsample size (i.e., the smallest value of k_{j}) were there is at least one gene with a survival analysis p-value <0.05 for each of the 1000 sampled datasets. Then, any gene with a p-value <0.05 in 80% of the 1000 sampled datasets would be called a GEAR for that cancer. The 80% value here is arbitrary but that is a minor point. I acknowledge that something must be chosen.

Presumably the gene with the largest effect for the cancer will define the value of K_{j} and, if the effect is large, this may result in other genes with smaller effects not being defined as a GEAR for that cancer by virtue of the 80% threshold. Thus, a gene being a GEAR is related to the strength of association for other genes in addition to its own strength of association. One could imagine that a gene that has a small-to-moderate effect consistently across many cancers may not show up as GEAR in any of them (if there are [potentially different] genes with more substantive effects for those cancers). Is this desirable?

The term "steadily associated" implies that a signal holds up across subsample gradients. Effectively this makes the subsampling a type of indirect adjustment to ensure the evidence of association is strong enough. How well this procedure performs in repeated use (i.e., as a statistical procedure) is not clear.

Assuredly subsampling sets the bar higher than requiring a nominal p-value to be beneath the 0.05 threshold based on analysis of the full data set. The author's note that the MEMORY has several methodological limitations, "chief among them is the need for rigorous, large-scale multiple-testing adjustment before any GEAR list can be considered clinically actionable." The reviewer agrees and would add that it may be difficult to address this limitation within the author's current framework. Moreover, should the author's method be used before such corrections are available given their statement? Perhaps clarification of what it means to be clinically actionable could help here. If a researcher uses MEMORY to screen for GEARs based on the current methodology, what do the authors recommend be done to select a subset of GEARs worthy of additional research/investment?

---

## [Referee Report · Reviewer #2 (Public review)]

Summary:

The authors are trying to come up with a list of genes (GEAR genes) that are consistently associated with cancer patient survival based on TCGA database. A method named "Multi-gradient Permutation Survival Analysis" was created based on bootstrapping and gradually increasing the sample size of the analysis. Only the genes with consistent performance in this analysis process are chosen as potential candidates for further analyses.

Strengths:

The authors describe in details their proposed method and the list of the chosen genes from the analysis. Scientific meaning and potential values of their findings are discussed in the context of published results in this field.

Weaknesses:

Some steps of the proposed method especially the definition survival analysis similarity (SAS) need further clarification or details since it would be difficult if anyone tries to reproduce the results.

If the authors can improve the clarity of the manuscript, including the proposed method and there is no major mistake there, the proposed approach can be applied to other diseases (assuming TCGA type of data is available for them) to identify potential gene lists, based on which drug screening can be performed to identify potential target for development.

---

## [Referee Report · Reviewer #4 (Public review)]

Thank you to the authors for their detailed responses and changes in relation to my questions. They have addressed all my concerns around methodological and inference clarity. I would still recommend against the use of feature/pathway selection techniques where there is no way of applying formal error control. I am pleased to read, however, that the authors are planning to develop this in future work. My edited review reflects these changes:

The authors apply what I gather is a novel methodology titled "Multi-gradient Permutation Survival Analysis" to identify genes that are robustly associated with prognosis ("GEARs") using tumour expression data from 15 cancer types available in the TCGA. The resulting lists of GEARs are then interrogated for biological insights using a range of techniques including connectivity and gene enrichment analysis.

I reviewed this paper primarily from a statistical perspective. Evidently an impressive amount of work has been conducted, concisely summarised, and great effort has been undertaken to add layers of insight to the findings. I am no stranger to what an undertaking this would have been. My primary concern, however, is that the novel statistical procedure proposed, and applied to identify the gene lists, as far as I can tell offers no statistical error control nor quantification. Consequently we have no sense what proportion of the highlighted GEAR genes and networks are likely to just be noise.

Major comments:

The main methodology used to identify the GEAR genes, "Multi-gradient Permutation Survival Analysis" does not formally account for multiple testing and offers no formal error control. Meaning we are left without knowing what the family wise (aka type 1) error rate is among the GEAR lists, nor the false discovery rate. I appreciate the emphasis on reproducibility, but I would generally recommend against the use of any feature selection methodology which does not provide error quantification because otherwise we do not know if we are encouraging our colleagues and/or readers to put resource into lists of genes that contain more noise than not. I am glad though and appreciative that the authors intend to develop this in future work.

The authors make a good point that, despite lack of validation in an external independent dataset, it is still compelling work given the functional characterisation and literature validation. I am pleased though that the authors agree validation in an independent dataset is an important next step, and plan to do so in future work.

---

## [Author Response]

The following is the authors’ response to the original reviews.

**Reviewer #1 (Public review):**
Summary:The authors propose a new technique which they name "Multi-gradient Permutation Survival Analysis (MEMORY)" that they use to identify "Genes Steadily Associated with Prognosis (GEARs)" using RNA-seq data from the TCGA database. The contribution of this method is one of the key stated aims of the paper. The vast majority of the paper focuses on various downstream analyses that make use of the specific GEARs identified by MEMORY to derive biological insights, with a particular focus on lung adenocarcinoma (LUAD) and breast invasive carcinoma (BRCA) which are stated to be representative of other cancers and are observed to have enriched mitosis and immune signatures, respectively. Through the lens of these cancers, these signatures are the focus of significant investigation in the paper.Strengths:The approach for MEMORY is well-defined and clearly presented, albeit briefly. This affords statisticians and bioinformaticians the ability to effectively scrutinize the proposed methodology and may lead to further advancements in this field.The scientific aspects of the paper (e.g., the results based on the use of MEMORY and the downstream bioinformatics workflows) are conveyed effectively and in a way that is digestible to an individual who is not deeply steeped in the cancer biology field.Weaknesses:I was surprised that comparatively little of the paper is devoted to the justification of MEMORY (i.e., the authors' method) for the identification of genes that are important broadly for the understanding of cancer. The authors' approach is explained in the methods section of the paper, but no rationale is given for why certain aspects of the method are defined as they are. Moreover, no comparison or reference is made to any other methods that have been developed for similar purposes and no results are shown to illustrate the robustness of the proposed method (e.g., is it sensitive to subtle changes in how it is implemented).For example, in the first part of the MEMORY algorithm, gene expression values are dichotomized at the sample median and a log-rank test is performed. This would seemingly result in an unnecessary loss of information for detecting an association between gene expression and survival. Moreover, while dichotomizing at the median is optimal from an information theory perspective (i.e., it creates equally sized groups), there is no reason to believe that median-dichotomization is correct vis-à-vis the relationship between gene expression and survival. If a gene really matters and expression only differentiates survival more towards the tail of the empirical gene expression distribution, median-dichotomization could dramatically lower the power to detect group-wise differences.

Thanks for these valuable comments!! We understand the reviewer’s concern regarding the potential loss of information caused by median-based dichotomization. In this study, we adopted the median as the cut-off value to stratify gene expression levels primarily for the purpose of data balancing and computational simplicity. This approach ensures approximately equal group sizes, which is particularly beneficial in the context of limited sample sizes and repeated sampling. While we acknowledge that this method may discard certain expression nuances, it remains a widely used strategy in survival analysis. To further evaluate and potentially enhance sensitivity, alternative strategies such as percentile-based cutoffs or survival models using continuous expression values (e.g., Cox regression) may be explored in future optimization of the MEMORY pipeline. Nevertheless, we believe that this dichotomization approach offers a straightforward and effective solution for the initial screening of survival-associated genes. We have now included this explanation in the revised manuscript (Lines 391–393).

Specifically, the authors' rationale for translating the Significant Probability Matrix into a set of GEARs warrants some discussion in the paper. If I understand correctly, for each cancer the authors propose to search for the smallest sample size (i.e., the smallest value of k_{j}) were there is at least one gene with a survival analysis p-value <0.05 for each of the 1000 sampled datasets. I base my understanding on the statement "We defined the sampling size k_{j} reached saturation when the max value of column j was equal to 1 in a significant-probability matrix. The least value of k_{j} was selected". Then, any gene with a p-value <0.05 in 80% of the 1000 sampled datasets would be called a GEAR for that cancer. The 80% value here seems arbitrary but that is a minor point. I acknowledge that something must be chosen. More importantly, do the authors believe this logic will work effectively in general? Presumably, the gene with the largest effect for a cancer will define the value of K_{j}, and, if the effect is large, this may result in other genes with smaller effects not being selected for that cancer by virtue of the 80% threshold. One could imagine that a gene that has a small-tomoderate effect consistently across many cancers may not show up as a gear broadly if there are genes with more substantive effects for most of the cancers investigated. I am taking the term "Steadily Associated" very literally here as I've constructed a hypothetical where the association is consistent across cancers but not extremely strong. If by "Steadily Associated" the authors really mean "Relatively Large Association", my argument would fall apart but then the definition of a GEAR would perhaps be suboptimal. In this latter case, the proposed approach seems like an indirect way to ensure there is a reasonable effect size for a gene's expression on survival.

Thank you for the comment and we apologize for the confusion! 𝐴_𝑖𝑗_ refers to the value of gene i under gradient j in the significant-probability matrix, primarily used to quantify the statistical probability of association with patient survival for ranking purposes. We believe that GEARs are among the top-ranked genes, but there is no established metric to define the optimal threshold. An 80% threshold is previously employed as an empirical standard in studies related to survival estimates [1]. In addition, we acknowledge that the determination of the saturation point 𝑘_𝑗_ is influenced by the earliest point at which any gene achieves consistent significance across 1000 permutations. We recognize that this may lead to the under representation of genes with moderate but consistent effects, especially in the presence of highly significant genes that dominate the statistical landscape. We therefore empirically used 𝐴_𝑖𝑗_ > 0.8 the threshold to distinguish between GEARs and non-GEARs. Of course, this parameter variation may indeed result in the loss of some GEARs or the inclusion of non-GEARs. We also agree that future studies could investigate alternative metrics and more refined thresholds to improve the application of GEARs.

Regarding the term ‘Steadily Associated’, we define GEARs based on statistical robustness across subsampled survival analyses within individual cancer types, rather than cross-cancer consistency or pan-cancer moderate effects. Therefore, our operational definition of “steadiness” emphasizes within-cancer reproducibility across sampling gradients, which does not necessarily exclude high-effect-size genes. Nonetheless, we agree that future extensions of MEMORY could incorporate cross-cancer consistency metrics to capture genes with smaller but reproducible pan-cancer effects.

The paper contains numerous post-hoc hypothesis tests, statements regarding detected associations and correlations, and statements regarding statistically significant findings based on analyses that would naturally only be conducted in light of positive results from analyses upstream in the overall workflow. Due to the number of statistical tests performed and the fact that the tests are sometimes performed using data-driven subgroups (e.g., the mitosis subgroups), it is highly likely that some of the findings in the work will not be replicable. Of course, this is exploratory science, and is to be expected that some findings won't replicate (the authors even call for further research into key findings). Nonetheless, I would encourage the authors to focus on the quantification of evidence regarding associations or claims (i.e., presenting effect estimates and uncertainty intervals), but to avoid the use of the term statistical significance owing to there being no clear plan to control type I error rates in any systematic way across the diverse analyses there were performed.

Thank you for the comment! We agree that rigorous control of type-I error is essential once a definitive list of prognostic genes is declared. The current implementation of MEMORY, however, is deliberately positioned as an exploratory screening tool: each gene is evaluated across 10 sampling gradients and 1,000 resamples per gradient, and the only quantity carried forward is its reproducibility probability (𝐴_𝑖𝑗_).

Because these probabilities are derived from aggregate “votes” rather than single-pass P-values, the influence of any one unadjusted test is inherently diluted. In another words, whether or not a per-iteration BH adjustment is applied does not materially affect the ranking of genes by reproducibility, which is the key output at this stage. However, we also recognize that a clinically actionable GEARs catalogue will require extensive, large-scale multiple-testing adjustments. Accordingly, future versions of MEMORY will embed a dedicated false-positive control framework tailored to the final GEARs list before any translational application. We have added this point in the ‘Discussion’ in the revised manuscript (Lines 350-359).

A prespecified analysis plan with hypotheses to be tested (to the extent this was already produced) and a document that defines the complete scope of the scientific endeavor (beyond that which is included in the paper) would strengthen the contribution by providing further context on the totality of the substantial work that has been done. For example, the focus on LUAD and BRCA due to their representativeness could be supplemented by additional information on other cancers that may have been investigated similarly but where results were not presented due to lack of space.

We thank the reviewer for requesting greater clarity on the analytic workflow. The MEMORY pipeline was fully specified before any results were examined and is described in ‘Methods’ (Lines 386–407). By contrast, the pathway-enrichment and downstream network/mutation analyses were deliberately exploratory: their exact content necessarily depended on which functional categories emerged from the unbiased GEAR screen.

Our screen revealed a pronounced enrichment of mitotic signatures in LUAD and immune signatures in BRCA.

We then chose these two cancer types for deeper “case-study” analysis because they contained the largest sample sizes among all cancers showing mitotic- or immune-dominated GEAR profiles, and provided the greatest statistical power for follow-up investigations. We have added this explanation into the revised manuscript (Line 163, 219-220).

**Reviewer #2 (Public review):**
Summary:The authors are trying to come up with a list of genes (GEAR genes) that are consistently associated with cancer patient survival based on TCGA database. A method named "Multi-gradient Permutation Survival Analysis" was created based on bootstrapping and gradually increasing the sample size of the analysis. Only the genes with consistent performance in this analysis process are chosen as potential candidates for further analyses.Strengths:The authors describe in detail their proposed method and the list of the chosen genes from the analysis. The scientific meaning and potential values of their findings are discussed in the context of published results in this field.Weaknesses:Some steps of the proposed method especially the definition of survival analysis similarity (SAS) need further clarification or details since it would be difficult if anyone tries to reproduce the results. In addition, the multiplicity (a large number of p-values are generated) needs to be discussed and/or the potential inflation of false findings needs to be part of the manuscript.

Thank you for the reviewer’s insightful comments. Accordingly, in the revised manuscript, we have provided a more detailed explanation of the definition and calculation of Survival-Analysis Similarity (SAS) to ensure methodological clarity and reproducibility (Lines 411-428); and the full code is now publicly available on GitHub (https://github.com/XinleiCai/MEMORY). We have also expanded the ‘Discussion’ to clarify our position on false-positive control: future releases of MEMORY will incorporate a dedicated framework to control false discoveries in the final GEARs catalogue, where itself will be subjected to rigorous, large-scale multiple-testing adjustment.

If the authors can improve the clarity of the proposed method and there is no major mistake there, the proposed approach can be applied to other diseases (assuming TCGA type of data is available for them) to identify potential gene lists, based on which drug screening can be performed to identify potential target for development.

Thank you for the suggestion. All source code has now been made publicly available on GitHub for reference and reuse. We agree that the GEAR lists produced by MEMORY hold considerable promise for drugscreening and target-validation efforts, and the framework could be applied to any disease with TCGA-type data. Of course, we also notice that the current GEAR catalogue should first undergo rigorous, large-scale multipletesting correction to further improve its precision before broader deployment.

**Reviewer #3 (Public review):**
Summary:The authors describe a valuable method to find gene sets that may correlate with a patient's survival. This method employs iterative tests of significance across randomised samples with a range of proportions of the original dataset. Those genes that show significance across a range of samples are chosen. Based on these gene sets, hub genes are determined from similarity scores.Strengths:MEMORY allows them to assess the correlation between a gene and patient prognosis using any available transcriptomic dataset. They present several follow-on analyses and compare the gene sets found to previous studies.Weaknesses:Unfortunately, the authors have not included sufficient details for others to reproduce this work or use the MEMORY algorithm to find future gene sets, nor to take the gene findings presented forward to be validated or used for future hypotheses.

Thank you for the reviewer’s comments! We apologize for the inconvenience and the lack of details.

Followed the reviewer’s valuable suggestion, we have now made all source code and relevant scripts publicly available on GitHub to ensure full reproducibility and facilitate future use of the MEMORY algorithm for gene discovery and hypothesis generation.

**Reviewer #4 (Public review):**
The authors apply what I gather is a novel methodology titled "Multi-gradient Permutation Survival Analysis" to identify genes that are robustly associated with prognosis ("GEARs") using tumour expression data from 15 cancer types available in the TCGA. The resulting lists of GEARs are then interrogated for biological insights using a range of techniques including connectivity and gene enrichment analysis.I reviewed this paper primarily from a statistical perspective. Evidently, an impressive amount of work has been conducted, and concisely summarised, and great effort has been undertaken to add layers of insight to the findings. I am no stranger to what an undertaking this would have been. My primary concern, however, is that the novel statistical procedure proposed, and applied to identify the gene lists, as far as I can tell offers no statistical error control or quantification. Consequently, we have no sense of what proportion of the highlighted GEAR genes and networks are likely to just be noise.Major comments:(1) The main methodology used to identify the GEAR genes, "Multi-gradient Permutation Survival Analysis" does not formally account for multiple testing and offers no formal error control. Meaning we are left with no understanding of what the family-wise (aka type 1) error rate is among the GEAR lists, nor the false discovery rate. I would generally recommend against the use of any feature selection methodology that does not provide some form of error quantification and/or control because otherwise we do not know if we are encouraging our colleagues and/or readers to put resources into lists of genes that contain more noise than not. There are numerous statistical techniques available these days that offer error control, including for lists of p-values from arbitrary sets of tests (see expansion on this and some review references below).

Thank you for your thoughtful and important comment! We fully agree that controlling type I error is critical when identifying gene sets for downstream interpretation or validation. As an exploratory study, our primary aim was to define and screen for GEARs by using the MEMORY framework; however, we acknowledge that the current implementation of MEMORY does not include a formal procedure for error control. Given that MEMORY relies on repeated sampling and counts the frequency of statistically significant p-values, applying standard p-value–based multiple-testing corrections at the individual test level would not meaningfully reduce the false-positive rate in this framework.

We believe that error control should instead be applied at the level of the final GEAR catalogue. However, we also recognize that conventional correction methods are not directly applicable. In future versions of MEMORY, we plan to incorporate a dedicated and statistically appropriate false-positive control module tailored specifically to the aggregated outputs of the pipeline. We have clarified this point explicitly in the revised manuscript. (Lines 350-359)

(2) Similarly, no formal significance measure was used to determine which of the strongest "SAS" connections to include as edges in the "Core Survival Network".

We agree that the edges in the Core Survival Network (CSN) were selected based on the top-ranked SAS values rather than formal statistical thresholds. This was a deliberate design choice, as the CSN was intended as a heuristic similarity network to prioritize genes for downstream molecular classification and biological exploration, not for formal inference. To address potential concerns, we have clarified this intent in the revised manuscript, and we now explicitly state that the network construction was based on empirical ranking rather than statistical significance (Lines 422-425).

(3) There is, as far as I could tell, no validation of any identified gene lists using an independent dataset external to the presently analysed TCGA data.

Thank you for the comment. We acknowledge that no independent external dataset was used in the present study to validate the GEARs lists. However, the primary aim of this work was to systematically identify and characterize genes with robust prognostic associations across cancer types using the MEMORY framework. To assess the biological relevance of the resulting GEARs, we conducted extensive downstream analyses including functional enrichment, mutation profiling, immune infiltration comparison, and drug-response correlation. These analyses were performed across multiple cancer types and further supported by a wide range of published literature.

We believe that this combination of functional characterization and literature validation provides strong initial support for the robustness and relevance of the GEARs lists. Nonetheless, we agree that validation in independent datasets is an important next step, and we plan to carry this out in future work to further strengthen the clinical application of MEMORY.

(4) There are quite a few places in the methods section where descriptions were not clear (e.g. elements of matrices referred to without defining what the columns and rows are), and I think it would be quite challenging to re-produce some aspects of the procedures as currently described (more detailed notes below).

We apologize for the confusion. In the revised manuscript, we have provided a clearer and more detailed description of the computational workflow of MEMORY to improve clarity and reproducibility.

(5) There is a general lack of statistical inference offered. For example, throughout the gene enrichment section of the results, I never saw it stated whether the pathways highlighted are enriched to a significant degree or not.

We apologize for not clearly stating this information in the original manuscript. In the revised manuscript, we have updated the figure legend to explicitly report the statistical significance of the enriched pathways (Line 870, 877, 879-880).

**Reviewer #1 (Recommendations for the authors):**
Overall, the paper reads well but there are numerous small grammatical errors that at times cost me non-trivial amounts of time to understand the authors' key messages.

We apologize for the grammatical errors that hindered clarity. In response, we have thoroughly revised the manuscript for grammar, spelling, and overall language quality.

**Reviewer #2 (Recommendations for the authors):**
Major comments:(1) Line 427: survival analysis similarity (SAS) definition. Any reference on this definition and why it is defined this way? Can the SAS value be negative? Based on line 429 definition, if A and B are exactly the same, SAS ~ 1; completely opposite, SAS = 0; otherwise, SAS could be any value, positive or negative. So it is hard to tell what SAS is measuring. It is important to make sure SAS can measure the similarity in a systematic and consistent way since it is used as input in the following network analysis.

We apologize for the confusion caused by the ambiguity in the original SAS formula. The SAS metric was inspired by the Jaccard index, but we modified the denominator to increase contrast between gene pairs. Specifically, the numerator counts the number of permutations in which both genes are simultaneously significant (i.e., both equal to 1), while the denominator is the sum of the total number of significant events for each gene minus twice the shared significant count. An additional +1 term was included in the denominator to avoid division by zero. This formulation ensures that SAS is always non-negative and bounded between 0 and 1, with higher values indicating greater similarity. We have clarified this definition and updated the formula in the revised manuscript (Lines 405-425).

(2) For the method with high dimensional data, multiplicity adjustment needs to be discussed, but it is missing in the manuscript. A 5% p-value cutoff was used across the paper, which seems to be too liberal in this type of analysis. The suggestion is to either use a lower cutoff value or use False Discovery Rate (FDR) control methods for such adjustment. This will reduce the length of the gene list and may help with a more focused discussion.

We appreciate the reviewer’s suggestion regarding multiplicity. MEMORY is intentionally positioned as an exploratory screen: each gene is tested across 10 sampling gradients and 1,000 resamples, and only its reproducibility probability (𝐴_𝑖𝑗_) is retained. Because this metric is an aggregate of 1,000 “votes” the influence of any single unadjusted P-value is already strongly diluted; adding a per-iteration BH/FDR step therefore has negligible impact on the reproducibility ranking that drives all downstream analyses.

That said, we recognize that a clinically actionable GEARs catalogue must undergo formal, large-scale multipletesting correction. Future releases of MEMORY will incorporate an error control module applied to the consolidated GEAR list before any translational use. We have now added a statement to this effect in the revised manuscript (Lines 350-359).

(3) To allow reproducibility from others, please include as many details as possible (software, parameters, modules etc.) for the analyses performed in different steps.

All source codes are now publically available on GitHub. We have also added the GitHub address in the section Online Content.

Minor comments or queries:(4) The manuscript needs to be polished to fix grammar, incomplete sentences, and missing figures.

Thank you for the suggestion. We have thoroughly proofread the manuscript to correct grammar, complete any unfinished sentences, and restore or renumber all missing figure panels. All figures are now properly referenced in the text.

(5) Line 131: "survival probability of certain genes" seems to be miss-leading. Are you talking about its probability of associating with survival (or prognosis)?

Sorry for the oversight. What we mean is the probability that a gene is found to be significantly associated with survival across the 1,000 resamples. We have revised the statement to “significant probability of certain genes” (Line 102).

(6) Lines 132, 133: "remained consistent": the score just needs to stay > 0.8 as the sample increases, or the score needs to be monotonously non-decreasing?

We mean the score stay above 0.8. We understand “remained consistent” is confusing and now revised it to “remained above 0.8”.

(7) Lines 168-170 how can supplementary figure 5A-K show "a certain degree of correlation with cancer stages"?

Sorry for the confusion! We have now revised Supplementary Figure 5A–K to support the visual impression with formal statistics. For each cancer type, we built a contingency table of AJCC stage (I–IV) versus hub-gene subgroup (Low, Mid, High) and applied Pearson’s 𝑥^2^ test (Monte-Carlo approximation, 10⁵ replicates when any expected cell count < 5). The 𝑥^2^ statistic and p-value are printed beneath every panel; eight of the eleven cancers show a significant association (p-value < 0.05), while LUSC, THCA and PAAD do not.We have replaced the vague phrase “a certain degree of correlation” with this explicit statistical statement in the revised manuscript (Lines 141-143).

(8) Lines 172-174: since the hub genes are a subset of GEAR genes through CSN construction, it is not a surprise of the consistency. any explanation about PAAD that is shown only in GOEA with GEARs but not with hub genes?

Thanks for raising this interesting point! In PAAD the Core Survival Network is unusually diffuse: the top-ranked SAS edges are distributed broadly rather than converging on a single dense module. Because of this flat topology, the ten highest-degree nodes (our hub set) do not form a tightly interconnected cluster, nor are they collectively enriched in the mitosis-related pathway that dominates the full GEAR list. This might explain that the mitotic enrichment is evident when all PAAD GEARs were analyzed but not when the analysis is confined to the far smaller—and more functionally dispersed—hub-gene subset.

(9) Lines 191: how the classification was performed? Tool? Cutoff values etc?

The hub-gene-based molecular classification was performed in R using hierarchical clustering. Briefly, we extracted the 𝑙𝑜𝑔_2_(𝑇𝑃𝑀 +1) expression matrix of hub genes, computed Euclidean distances between samples, and applied Ward’s minimum variance method (hclust, method = "ward.D2"). The resulting dendrogram was then divided into three groups (cutree, k = 3), corresponding to low, mid, and high expression classes. These parameters were selected based on visual inspection of clustering structure across cancer types. We have added this information to the revised ‘Methods’ section (Lines 439-443).

(10) Lines 210-212: any statistics to support the conclusion? The bar chat of Figure 3B seems to support that all mutations favor ML & MM.

We agree that formal statistical support is important for interpreting groupwise comparisons. In this case, however, several of the driver events, such as ROS1 and ERBB2, had very small subgroup counts, which violate the assumptions of Pearson’s 𝑥^2^ test. While we explored 𝑥^2^ and Fisher’s exact tests, the results were unstable due to sparse counts. Therefore, we chose to present these distributions descriptively to illustrate the observed subtype preferences across different driver mutations (Figure 3B). We have revised the manuscript text to clarify this point (Lines 182-188).

(11) Line 216: should supplementary Figure 6H-J be "6H-I"?

We apologize for the mistake. We have corrected it in the revised manuscript.

(12) Line 224: incomplete sentence starting with "To further the functional... ".

Thanks! We have made the revision and it states now “To further expore the functional implications of these mutations, we enriched them using a pathway system called Nested Systems in Tumors (NeST)”.

(13) Lines 261-263: it is better to report the median instead of the mean. Use log scale data for analysis or use non-parametric methods due to the long tail of the data.

Thank you for the very helpful suggestion. In the revised manuscript, we now report the median instead of the mean to better reflect the distribution of the data. In addition, we have applied log-scale transformation where appropriate and replaced the original statistical tests with non-parametric Wilcoxon ranksum tests to account for the long-tailed distribution. These changes have been implemented in both the main text and figure legends (Lines 234–237, Figure 5F).

(14) Line 430: why based on the first sampling gradient, i.e. k_1 instead of the k_j selected? Or do you mean k_j here?

Thanks for this question! We deliberately based SAS on the vectors from the first sampling gradient (𝑘_1_, ≈ 10 % of the cohort). At this smallest sample size, the binary significance patterns still contain substantial variation, and many genes are not significant in every permutation. Based on this, we think the measure can meaningfully identify gene pairs that behave concordantly throughout the gradient permutation.

We have now added a sentence to clarify this in the Methods section (Lines 398–403).

(15) Need clarification on how the significant survival network was built.

Thank you for pointing this out. We have now provided a more detailed clarification of how the Survival-Analysis Similarity (SAS) metric was defined and applied in constructing the core survival network (CSN), including the rationale for key parameter choices (Lines 409–430). Additionally, we have made full source code publicly available on GitHub to facilitate transparency and reproducibility (https://github.com/XinleiCai/MEMORY).

(16) Line 433: what defines the "significant genes" here? Are they the same as GEAR genes? And what are total genes, all the genes?

We apologize for the inconsistency in terminology, which may have caused confusion. In this context,

“significant genes” refers specifically to the GEARs (Genes Steadily Associated with Prognosis). The SAS values were calculated between each GEAR and all genes. We have revised the manuscript to clarify this by consistently using the term “GEARs” throughout.

(17) Line 433: more detail on how SAS values were used will be helpful. For example, were pairwise SAS values fed into Cytoscape as an additional data attribute (on top of what is available in TCGA) or as the only data attribute for network building?

The SAS values were used as the sole metric for defining connections (edges) between genes in the construction of the core survival network (CSN). Specifically, we calculated pairwise SAS values between each GEAR and all other genes, then selected the top 1,000 gene pairs with the highest SAS scores to construct the network. No additional data attributes from TCGA (such as expression levels or clinical features) were used in this step. These selected pairs were imported into Cytoscape solely based on their SAS values to visualize the CSN.

(18) Line 434: what is "ranking" here, by degree? Is it the same as "nodes with top 10 degrees" at line 436?

The “ranking” refers specifically to the SAS values between gene pairs. The top 1,000 ranked SAS values were selected to define the edges used in constructing the Core Survival Network (CSN).

Once the CSN was built, we calculated the degree (number of connections) for each node (i.e., each gene). The

“top 10 degrees” mentioned on Line 421 refers to the 10 genes with the highest node degrees in the CSN. These were designated as hub genes for downstream analyses.

We have clarified this distinction in the revised manuscript (Line 398-403).

(19) Line 435: was the network built in Cytoscape? Or built with other tool first and then visualized in Cytoscape?

The network was constructed in R by selecting the top 1,000 gene pairs with the highest SAS values to define the edges. This edge list was then imported into Cytoscape solely for visualization purposes. No network construction or filtering was performed within Cytoscape itself. We have clarified this in the revised ‘Methods’ section (Lines 424-425).

(20) Line 436: the degree of each note was calculated, what does it mean by "degree" here and is it the same as the number of edges? How does it link to the "higher ranked edges" in Line 165?

The “degree” of a node refers to the number of edges connected to that node—a standard metric in graph theory used to quantify a node’s centrality or connectivity in the network. It is equivalent to the number of edges a gene shares with others in the CSN.

The “higher-ranked edges” refer to the top 1,000 gene pairs with the highest SAS values, which we used to construct the Core Survival Network (CSN). The degree for each node was computed within this fixed network, and the top 10 nodes with the highest degree were selected as hub genes. Therefore, the node degree is largely determined by this pre-defined edge set.

(21) Line 439: does it mean only 1000 SAS values were used or SAS values from 1000 genes, which should come up with 1000 choose 2 pairs (~ half million SAS values).

We computed the SAS values between each GEAR gene and all other genes, resulting in a large number of pairwise similarity scores. Among these, we selected the top 1,000 gene pairs with the highest SAS values—regardless of how many unique genes were involved—to define the edges in the Core Survival Network (CSN). In another words, the network is constructed from the top 1,000 SAS-ranked gene pairs, not from all possible combinations among 1,000 genes (which would result in nearly half a million pairs). This approach yields a sparse network focused on the strongest co-prognostic relationships.

We have clarified this in the revised ‘Methods’ section (Lines 409–430).

(22) Line 496: what tool is used and what are the parameters set for hierarchical clustering if someone would like to reproduce the result?

The hierarchical clustering was performed in R using the hclust function with Ward's minimum variance method (method = "ward.D2"), based on Euclidean distance computed from the log-transformed expression matrix (𝑙𝑜𝑔_2_(𝑇𝑃𝑀 +1)). Cluster assignment was done using the cutree function with k = 3 to define low, mid, and high expression subgroups. These settings have now been explicitly stated in the revised ‘Methods’ section (Lines 439–443) to facilitate reproducibility.

(23) Lines 901-909: Figure 4 missing panel C. Current panel C seems to be the panel D in the description.

Sorry for the oversights and we have now made the correction (Line 893).

(24) Lines 920-928: Figure 6C: considering a higher bar to define "significant".

We agree that applying a more stringent cutoff (e.g., p < 0.01) may reduce potential false positives. However, given the exploratory nature of this study, we believe the current threshold remains appropriate for the purpose of hypothesis generation.

**Reviewer #3 (Recommendations for the authors):**
(1) The title says the genes that are "steadily" associated are identified, but what you mean by the word "steadily" is not defined in the manuscript. Perhaps this could mean that they are consistently associated in different analyses, but multiple analyses are not compared.

In our manuscript, “steadily associated” refers to genes that consistently show significant associations with patient prognosis across multiple sample sizes and repeated resampling within the MEMORY framework (Lines 65–66). Specifically, each gene is evaluated across 10 sampling gradients (from ~10% to 100% of the cohort) with 1,000 permutations at each level. A gene is defined as a GEAR if its probability of being significantly associated with survival remains ≥ 0.8 throughout the whole permutation process. This stability in signal under extensive resampling is what we refer to as “steadily associated.”

(2) I think the word "gradient" is not appropriately used as it usually indicates a slope or a rate of change. It seems to indicate a step in the algorithm associated with a sampling proportion.

Thank you for pointing out the potential ambiguity in our use of the term “gradient.” In our study, we used “gradient” to refer to stepwise increases in the sample proportion used for resampling and analysis. We have now revised it to “progressive”.

(3) Make it clear that the name "GEARs" is introduced in this publication.

Done.

(4) Sometimes the document is hard to understand, for example, the sentence, "As the number of samples increases, the survival probability of certain genes gradually approaches 1." It does not appear to be calculating "gene survival probability" but rather a gene's association with patient survival. Or is it that as the algorithm progresses genes are discarded and therefore do have a survival probability? It is not clear.

What we intended to describe is the probability that a gene is judged significant in the 1,000 resamples at a given sample-size step, that is, its reproducibility probability in the MEMORY framework. We have now revised the description (Lines 101-104).

(5) The article lacks significant details, like the type of test used to generate p-values. I assume it is the log-rank test from the R survival package. This should be explicitly stated. It is not clear why the survminer R package is required or what function it has. Are the p-values corrected for multiple hypothesis testing at each sampling?

We apologize for the lack of details. In each sampling iteration, we used the log-rank test (implemented via the survdiff function in the R survival package) to evaluate the prognostic association of individual genes. This information has now been explicitly added to the revised manuscript.

The survminer package was originally included for visualization purposes, such as plotting illustrative Kaplan– Meier curves. However, since it did not contribute to the core statistical analysis, we have now removed this package from the Methods section to avoid confusion (Lines 386-407).

As for multiple-testing correction, we did not adjust p-values in each iteration, because the final selection of GEARs is based on the frequency with which a gene is found significant across 1,000 resamples (i.e., its reproducibility probability). Classical FDR corrections at the per-sample level do not meaningfully affect this aggregate metric. That said, we fully acknowledge the importance of multiple-testing control for the final GEARs catalogue. Future versions of the MEMORY framework will incorporate appropriate adjustment procedures at that stage.

(6) It is not clear what the survival metric is. Is it overall survival (OS) or progression-free survival (PFS), which would be common choices?

It’s overall survival (OS).

(7) The treatment of the patients is never considered, nor whether the sequencing was performed pre or posttreatment. The patient's survival will be impacted by the treatment that they receive, and many other factors like commodities, not just the genomics.

We initially thought there exist no genes significantly associated with patient survival (GEARs) without counting so many different influential factors. This is exactly what motivated us to invent the

MEMORY. However, this work proves “we were wrong”, and it demonstrates the real power of GEARs in determining patient survival. Of course, we totally agree with the reviewer that incorporating therapy variables and other clinical covariates will further improve the power of MEMORY analyses.

(8) As a paper that introduces a new analysis method, it should contain some comparison with existing state of the art, or perhaps randomised data.

Our understanding is --- the MEMORY presents as an exploratory and proof-of-concept framework. Comparison with regular survival analyses seems not reasonable. We have added some discussion in revised manuscript (Lines 350-359).

(9) In the discussion it reads, "it remains uncertain whether there exists a set of genes steadily associated with cancer prognosis, regardless of sample size and other factors." Of course, there are many other factors that may alter the consistency of important cancer genes, but sample size is not one of them. Sample size merely determines whether your study has sufficient power to detect certain gene effects, it does not effect whether genes are steadily associated with cancer prognosis in different analyses. (Of course, this does depend on what you mean by "steadily".)

We totally agree with reviewer that sample size itself does not alter a gene’s biological association with prognosis; it only affects the statistical power to detect that association. Because this study is exploratory and we were initially uncertain whether GEARs existed, we first examined the impact of sample-size variation—a dominant yet experimentally tractable source of heterogeneity—before considering other, less controllable factors.

**Reviewer #4 (Recommendations for the authors):**
Other more detailed comments:(1) IntroductionL93: When listing reasons why genes do not replicate across different cohorts / datasets, there is also the simple fact that some could be false positives

We totally agree that some genes may simply represent false-positive findings apart from biological heterogeneity and technical differences between cohorts. Although the MEMORY framework reduces this risk by requiring high reproducibility across 1,000 resamples and multiple sample-size tiers, it cannot eliminate false positives completely. We have added some discussion and explicitly note that external validation in independent datasets is essential for confirming any GEAR before clinical application.

(2) Results SectionL143: Language like "We also identified the most significant GEARs in individual cancer types" I think is potentially misleading since the "GEAR" lists do not have formal statistical significance attached.

We removed “significant” ad revised it to “top 1” (Line 115).

L153 onward: The pathway analysis results reported do not include any measures of how statistically significant the enrichment was.

We have now updated the figure legends to clearly indicate that the displayed pathways represent the top significantly enriched results based on adjusted p-values from GO enrichment analyses (Lines 876-878).

L168: "A certain degree of correlation with cancer stages (TNM stages) is observed in most cancer types except for COAD, LUSC and PRAD". For statements like this statistical significance should be mentioned in the same sentence or, if these correlations failed to reach significance, that should be explicitly stated.

In the revised Supplementary Figure 5A–K, we now accompany the visual trends with formal statistical testing. Specifically, for each cancer type, we constructed a contingency table of AJCC stage (I–IV) versus hub-gene subgroup (Low, Mid, High) and applied Pearson’s 𝑥^2^ test (using Monte Carlo approximation with 10⁵ replicates if any expected cell count was < 5). The resulting 𝑥^2^ statistic and p-value are printed beneath each panel. Of the eleven cancer types analyzed, eight showed statistically significant associations (p < 0.05), while COAD, LUSC, and PRAD did not. Accordingly, we have make the revision in the manuscript (Line 137139).

L171-176: When mentioning which pathways are enriched among the gene lists, please clarify whether these levels of enrichment are statistically significant or not. If the enrichment is significant, please indicate to what degree, and if not I would not mention.

We agree that the statistical significance of pathway enrichment should be clearly stated and made the revision throughout the manuscript (Line 869, 875, 877).

(3) Methods SectionL406 - 418: I did not really understand, nor see it explained, what is the motivation and value of cycling through 10%, 20% bootstrapped proportions of patients in the "gradient" approach? I did not see this justified, or motivated by any pre-existing statistical methodology/results. I do not follow the benefit compared to just doing one analysis of all available samples, and using the statistical inference we get "for free" from the survival analysis p-values to quantify sampling uncertainty.

The ten step-wise sample fractions (10 % to 100 %) allow us to transform each gene’s single log-rank P-value into a reproducibility probability: at every fraction we repeat the test 1,000 times and record the proportion of permutations in which the gene is significant. This learning-curve-style resampling not only quantifies how consistently a gene associates with survival under different power conditions but also produces the 0/1 vectors required to compute Survival-Analysis Similarity (SAS) and build the Core Survival Network. A single one-off analysis on the full cohort would yield only one P-value per gene, providing no binary vectors at all—hence no basis for calculating SAS or constructing the network.

L417: I assume p < 0.05 in the survival analysis means the nominal p-value, unadjusted for multiple testing. Since we are in the context of many tests please explicitly state if so.

Yes, p < 0.05 refers to the nominal, unadjusted p-value from each log-rank test within a single permutation. In MEMORY these raw p-values are converted immediately into 0/1 “votes” and aggregated over 1 000 permutations and ten sample-size tiers; only the resulting reproducibility probability (𝐴_𝑖𝑗_) is carried forward. No multiple-testing adjustment is applied at the individual-test level, because a per-iteration FDR or BH step would not materially affect the final 𝐴_𝑖𝑗_ ranking. We have revised the manuscript (Line 396)

L419-426: I did not see defined what the rows are and what the columns are in the "significant-probability matrix". Are rows genes, columns cancer types? Consequently I was not really sure what actually makes a "GEAR". Is it achieving a significance probability of 0.8 across all 15 cancer subtypes? Or in just one of the tumour datasets?

In the significant-probability matrix, each row represents a gene, and each column corresponds to a sampling gradient (i.e., increasing sample-size tiers from ~10% to 100%) within a single cancer type. The matrix is constructed independently for each cancer.

GEAR is defined as achieving a significance probability of 0.8 within a single tumor type. Not need to achieve significance probability across all 15 cancer subtypes.

L426: The significance probability threshold of 0.8 across 1,000 bootstrapped nominal tests --- used to define the GEAR lists --- has, as far as I can tell, no formal justification. Conceptually, the "significance probability" reflects uncertainty in the patients being used (if I follow their procedure correctly), but as mentioned above, a classical p-value is also designed to reflect sampling uncertainty. So why use the bootstrapping at all?Moreover, the 0.8 threshold is applied on a per-gene basis, so there is no apparent procedure "built in" to adapt to (and account for) different total numbers of genes being tested. Can the authors quantify the false discovery rate associated with this GEAR selection procedure e.g. by running for data with permuted outcome labels? And why do the gradient / bootstrapping at all --- why not just run the nominal survival p-values through a simple Benjamini-Hochberg procedure, and then apply and FDR threshold to define the GEAR lists? Then you would have both multiplicity and error control for the final lists. As it stands, with no form of error control or quantification of noise rates in the GEAR lists I would not recommend promoting their use. There is a long history of variable selection techniques, and various options the authors could have used that would have provided formal error rates for the final GEAR lists (see seminal reviews by eg Heinze et al 2018) BiometricalJournal, or O'Hara and Sillanpaa, 2009, Bayesian Analysis, including, as I say, simple application of a Benjamini-Hochberg to achive multiplicity adjusted FDR control.

Thank you. We chose the 10 × 1,000 resampling scheme to ask a different question from a single Benjamini–Hochberg scan: does a gene keep re-appearing as significant when cohort composition and statistical power vary from 10 % to 100 % of the data? Converting the 1,000 nominal p-values at each sample fraction into a reproducibility probability 𝐴_𝑖𝑗_ allows us to screen for signals that are stable across wide sampling uncertainty rather than relying on one pass through the full cohort. The 0.8 cut-off is an intentionally strict, empirically accepted robustness threshold (analogous to stability-selection); under the global null the chance of exceeding it in 1,000 draws is effectively zero, so the procedure is already highly conservative even before any gene-wise multiplicity correction [1]. Once MEMORY moves beyond this exploratory stage and a final, clinically actionable GEAR catalogue is required, we will add a formal FDR layer after the robustness screen, but for the present proof-of-concept study, we retain the resampling step specifically to capture stability rather than to serve as definitive error control.

L427-433: I gathered that SAS reflects, for a particular pair of genes, how likely they are to be jointly significant across bootstraps. If so, perhaps this description or similar could be added since I found a "conceptual" description lacking which would have helped when reading through the maths. Does it make sense to also reflect joint significance across multiple cancer types in the SAS? Or did I miss it and this is already reflected?

SAS is indeed meant to quantify, within a single cancer type, how consistently two genes are jointly significant across the 1,000 bootstrap resamples performed at a given sample-size tier. In another words, SAS is the empirical probability that the two genes “co-light-up” in the same permutation, providing a measure of shared prognostic behavior beyond what either gene shows alone. We have added this plain language description to the ‘Methods’ (Lines 405-418).

In the current implementation SAS is calculated separately for each cancer type; it does not aggregate cosignificance across different cancers. Extending SAS to capture joint reproducibility across multiple tumor types is an interesting idea, especially for identifying pan-cancer gene pairs, and we note this as a potential future enhancement of the MEMORY pipeline.

L432: "The SAS of significant genes with total genes was calculated, and the significant survival network was constructed" Are the "significant genes" the "GEAR" list extracted above according to the 0.8 threshold? If so, and this is a bit pedantic, I do not think they should be referred to as "significant genes" and that this phrase should be reserved for formal statistical significance.

We have replaced “significant genes” with “GEAR genes” to avoid any confusion (Lines 421-422).

L434: "some SAS values at the top of the rankings were extracted, and the SAS was visualized to a network by Cytoscape. The network was named core survival network (CSN)". I did not see it explicitly stated which nodes actually go into the CSN. The entire GEAR list? What threshold is applied to SAS values in order to determine which edges to include? How was that threshold chosen? Was it data driven? For readers not familiar with what Cytoscape is and how it works could you offer more of an explanation in-text please? I gather it is simply a piece of network visualisation/wrangling software and does not annotate additional information (e.g. external experimental data), which I think is an important point to clarify in the article without needing to look up the reference.

We have now clarified these points in the revised ‘Methods’ section, including how the SAS threshold was selected and which nodes were included in the Core Survival Network (CSN). Specifically, the CSN was constructed using the top 1,000 gene pairs with the highest SAS values. This threshold was not determined by a fixed numerical cutoff, but rather chosen empirically after comparing networks built with varying numbers of edges (250, 500, 1,000, 2,000, 6,000, and 8,000; see Reviewer-only Figure 1). We observed that, while increasing the number of edges led to denser networks, the set of hub genes remained largely stable. Therefore, we selected 1,000 edges as a balanced compromise between capturing sufficient biological information and maintaining computational efficiency and interpretability.

The resulting node list (i.e., the genes present in those top-ranked pairs) is provided in Supplementary Table 4. Cytoscape was used solely as a network visualization platform, and no external annotations or experimental data were added at this stage. We have added a brief clarification in the main text to help readers understand.

L437: "The effect of molecular classification by hub genes is indicated that 1000 to 2000 was a range that the result of molecular classification was best." Can you clarify how "best" is assessed here, i.e. by what metric and with which data?

We apologize for the confusion. Upon constructing the network, we observed that the number of edges affected both the selection of hub genes and the computational complexity. We analyzed the networks with 250, 500, 1,000, 2,000, 6,000 and 8,000 edges, and found that the differences in selected hub genes were small (Author response image 1). Although the networks with fewer edges had lower computational complexity, the choice of 1000 edges was a compromise to the balance between sufficient biological information and manageable computational complexity. Thus, we chose the network with 1,000 edges as it offered a practical balance between computational efficiency and the biological relevance of the hub genes.

**Author response image 1. sa4fig1:** The intersection of the network constructed by various number of edges.

References

(1) Gebski, V., Garès, V., Gibbs, E. & Byth, K. Data maturity and follow-up in time-to-event analyses.International Journal of Epidemiology 47, 850–859 (2018).